# A single amino acid substitution in the capsid protein of Zika virus contributes to a neurovirulent phenotype

Guang-Yuan Song[1,2,6], Xing-Yao Huang[2,6], Meng-Jiao He[2,6], Hang-Yu Zhou[3,6], Rui-Ting Li[2], Ying Tian[1,2], Yan Wang [4], Meng-Li Cheng[2], Xiang Chen[2], Rong-Rong Zhang[2], Chao Zhou [2], Jia Zhou [2], Xian-Yang Fang [5], Xiao-Feng Li[2] ✉ & Cheng-Feng Qin [1,2] ✉

Increasing evidence shows the African lineage Zika virus (ZIKV) displays a more severe neurovirulence compared to the Asian ZIKV. However, viral determinants and the underlying mechanisms of enhanced virulence phenotype remain largely unknown. Herein, we identify a panel of amino acid substitutions that are unique to the African lineage of ZIKVs compared to the Asian lineage by phylogenetic analysis and sequence alignment. We then utilize reverse genetic technology to generate recombinant ZIKVs incorporating these lineage-specific substitutions based on an infectious cDNA clone of Asian ZIKV. Through in vitro characterization, we discover a mutant virus with a lysine to arginine substitution at position 101 of capsid (C) protein (termed K101R) displays a larger plaque phenotype, and replicates more efficiently in various cell lines. Moreover, K101R replicates more efficiently in mouse brains and induces stronger inflammatory responses than the wild type (WT) virus in neonatal mice. Finally, a combined analysis reveals the K101R substitution promotes the production of mature C protein without affecting its binding to viral RNA. Our study identifies the role of K101R substitution in the C protein in contributing to the enhanced virulent phenotype of the African lineage ZIKV, which expands our understanding of the complexity of ZIKV proteins.

The mosquito-transmitted Zika virus (ZIKV) is a member of *Flavivirus* genus in the family *Flaviviridae*, which includes a number of well-known human pathogens such as yellow fever virus (YFV), dengue virus (DENV), West Nile virus (WNV), and Japanese encephalitis virus (JEV). Originally isolated in 1947 in the Zika forest of Uganda, Africa[1], ZIKV was neglected for many years, as only sporadic human infections were documented in Africa and Asia[2,3]. It is noteworthy, however, during the 2015–2016 outbreak in the Americas, ZIKV spread into 84 different countries worldwide, with an estimated total of over 1.5 million cases, including thousands of microcephaly cases and other congenital malformations now termed congenital Zika syndrome (CZS)[4,5]. On February 1st, 2016, the World Health Organization (WHO) declared ZIKV outbreaks a public health emergency of international concern. Although ZIKV transmission has waned in the Americas,

[1]School of Basic Medical Sciences, Anhui Medical University, 230032 Hefei, Anhui, China. [2]Department of Virology, State Key Laboratory of Pathogen and Biosecurity, Beijing Institute of Microbiology and Epidemiology, Academy of Military Medical Sciences, 100071 Beijing, China. [3]Suzhou Institute of System Medicine, Chinese Academy of Medical Sciences & Peking Union Medical College, 215123 Suzhou, Jiangsu, China. [4]Beijing Advanced Innovation Center for Structural Biology, School of Life Sciences, Tsinghua University, 100084 Beijing, China. [5]Key Laboratory of RNA Biology, Institute of Biophysics, Chinese Academy of Sciences, 100101 Beijing, China. [6]These authors contributed equally: Guang-Yuan Song, Xing-Yao Huang, Meng-Jiao He, Hang-Yu Zhou. ✉e-mail: xiaofeng_li_bj@163.com; qincf@bmi.ac.cn

outbreaks and infection clusters continue to emerge in Asia, India, and Africa[6]. Currently, no antiviral agents and vaccines have been approved for use in preventing and treating ZIKV infection.

The ZIKV genome is a positive-sense single-stranded RNA, which consists of a single open reading frame (ORF) flanked by 5' and 3' untranslated regions (UTR). The ORF produces a polyprotein, which is subsequently processed by viral and host proteases to produce a total of three structural proteins (capsid (C), pre-membrane/membrane (prM/M), and envelope proteins (E)) and seven nonstructural proteins (NS1, NS2A, NS2B, NS3, NS4A, NS4B, and NS5). The structural proteins are responsible for virus assembly, while the nonstructural proteins are involved in the replication of the viral RNA genome. Phylogenetic analysis shows that ZIKV strains are classified into two major lineages referred to as the ancestral African lineage and the contemporary Asian lineage, respectively[7]. To date, the African lineage has exclusively been detected in the African continent[8,9], whereas the majority of recent epidemics in the South Pacific and Americas are attributed to the Asian lineage[10,11]. Contemporary Asian lineage ZIKV has acquired multiple adaptive mutations during its circulation to enhance its transmissibility from mosquitoes to humans[12] and tropism to human neural progenitor cells (NPC)[13]. Significantly, comparative studies based on cell culture experiments and animal models have shown that African lineage viruses were more virulent than Asian lineage viruses[14–19]. For example, when compared to Asian strains, African strains infected human NSCs and astrocytes more efficiently, and caused greater levels of apoptosis[20,21]. Furthermore, African ZIKV isolates caused more rapid and more severe cytopathic effects (CPE) and replicated more efficiently than Asian isolates in multiple cell lines[22,23]. ZIKV structural genes have been reported to determine the virulence of African and Asian lineages[24]. Nevertheless, viral determinants and underlying mechanisms responsible for the phenotypic differences remain largely unknown currently.

Here, combining reverse genetics, animal experiments, and biochemical assays, we aimed to identify the molecular neurovirulent determinants of the African lineage. Our findings revealed that substitution from lysine to arginine at position 101 of the C protein (termed K101R) contributed to the enhanced virulence phenotype in both cell culture and animal models.

## Results

### Virulence phenotype and amino acid differences between structural proteins of African and Asian ZIKV lineages

To make a direct comparison of the virulence phenotype between the Asian and African ZIKV strains, we selected the African prototype strain MR766 and the Asian representative FSS13025 strain for analysis. As shown in Fig. S1a and S1b, the MR766 strain replicated more efficiently and caused more severe CPE in Vero cells compared to the FSS13025 strain. Furthermore, intracranial (i.c.) inoculation of CD-1 neonatal mice with 1 PFU of MR766 led to 100% mortality, whereas the same dosage of FSS13025 resulted in only 10% mortality (Fig. S1c). These findings are consistent with previous studies reporting that the African ZIKV is more virulent than the Asian ZIKV in cell cultures and neonatal mouse models[14–19].

Due to the African prototype strain MR766 has been widely used to compare the divergence between two distinct lineages, to identify the potential functional residues responsible for this lineage-specific phenotype, we further performed phylogenetic analysis and sequence alignment of the ZIKV genomes available in GenBank. As expected, ZIKV strains were phylogenetically grouped into two distinct branches: the African and Asian lineages (Fig. 1a). Then, sequence alignment revealed a total of 11 lineage-specific residues in the structural proteins of ZIKV, which were located at positions 101, 110, and 125 in the C protein, positions 148, 246, and 262 in the prM protein, and positions 410, 459, 607, 728, and 785 in the E protein, respectively (Fig. S2). Six

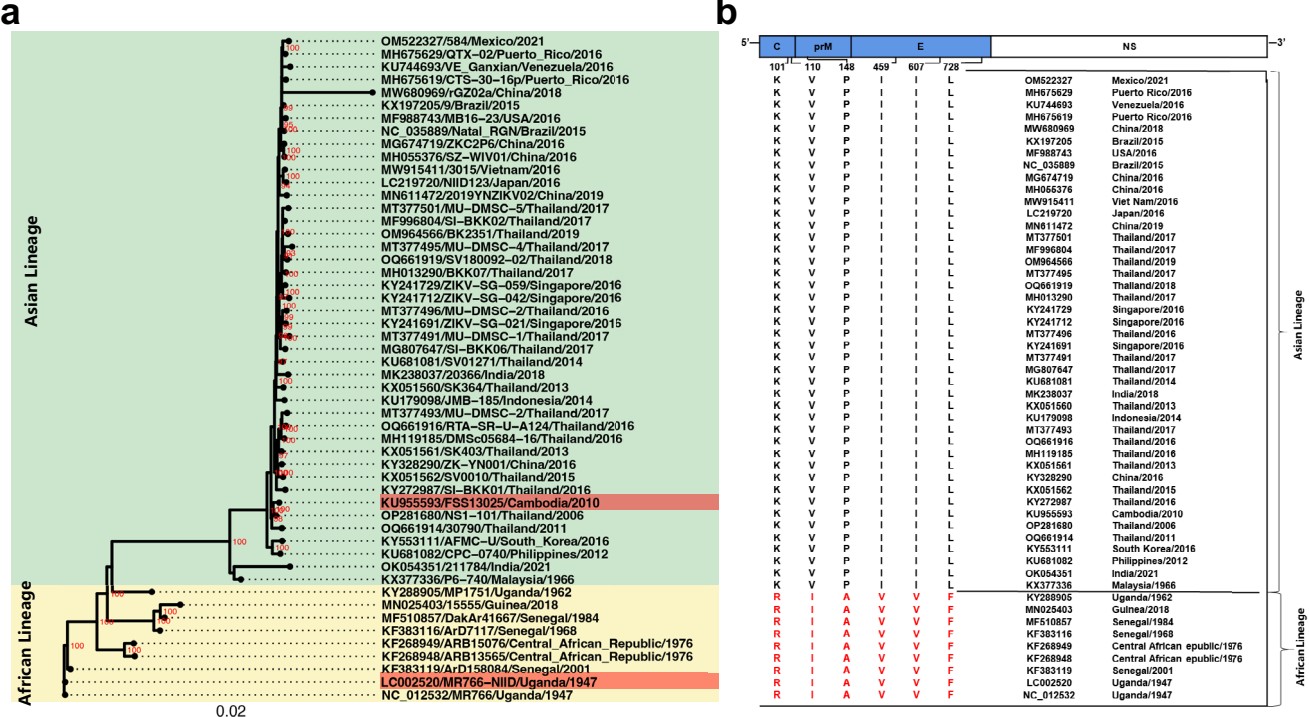

**Fig. 1 | Phylogenetic analysis of African and Asian ZIKVs. a** The maximum likelihood phylogenetic tree of 52 ZIKV complete sequences was constructed using IQ-Tree (v1.6.12) with 1000 ultrafast bootstrap test. GTR + F + G4 substitution model was chosen as the Best-fit model according to Bayesian Information Criterion (BIC). The bootstrap value of tree nodes was labeled beside the node if it higher than 80. Strains used in this study were labeled in red. The African lineages and Asian lineages were highlighted with different background colors. **b** Variations among ZIKV strains between the Asian lineage and African lineage were identified by sequences alignment with MAFFT v7.4.1 and the potential virulent sites were listed.

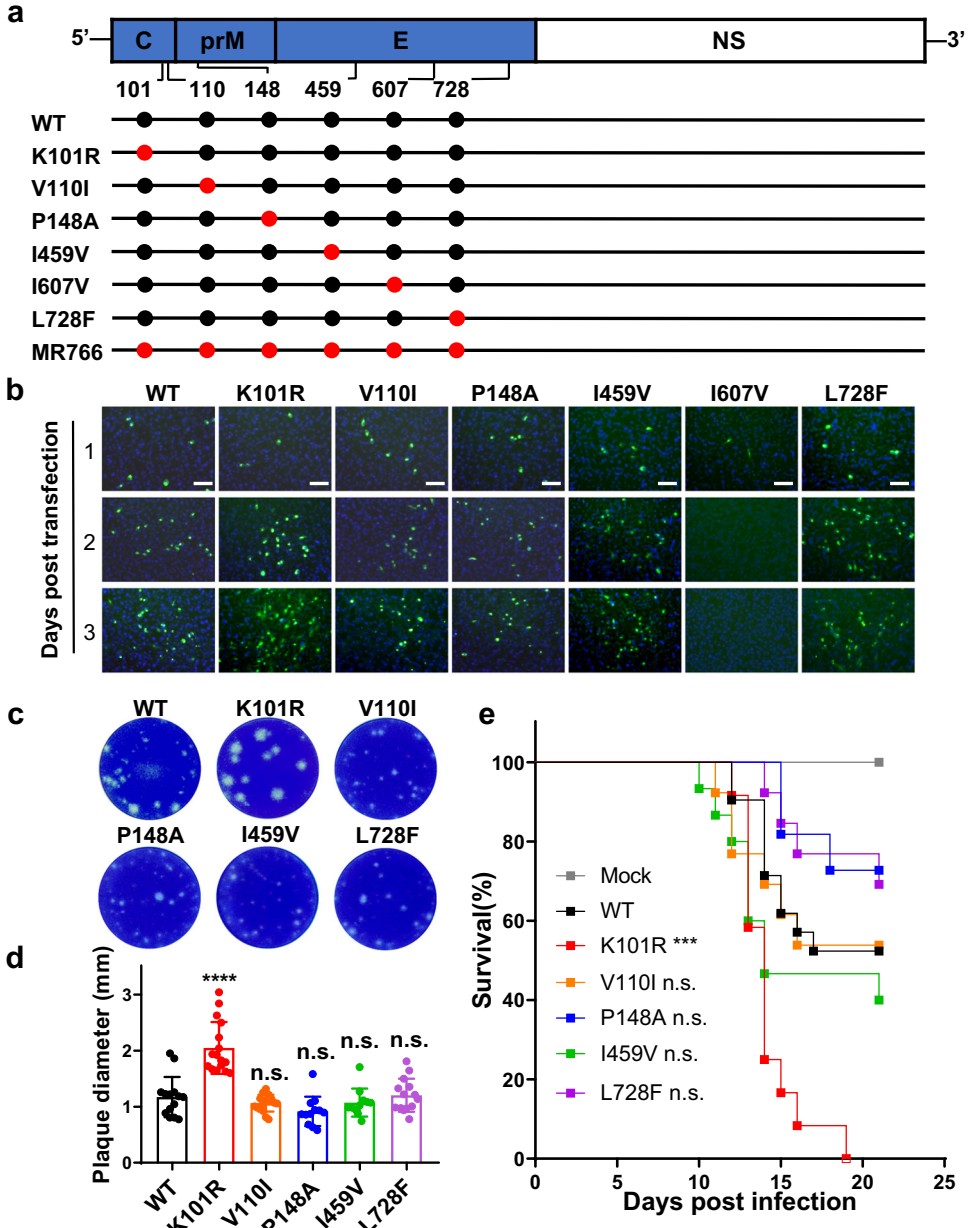

**Fig. 2 | Design and characterization of recombinant ZIKVs with African lineage-specific substitutions. a** Schematic of genomes of WT, mutant ZIKVs, and MR766. Locations of substitutions are indicated by red dots in the diagrams of the ZIKV genome. **b** Immunofluorescent assay (IFA) of BHK-21 cells transfected with equal amounts of RNAs of recombinant viruses. E protein expression was detected on days 1 to 3 post-transfection. Scale bars: 100 μm. **c** Plaque morphology of recombinant ZIKV mutant viruses in BHK-21 cells. **d** Measurement of plaque size in BHK-21 cells infected with indicated viruses. WT: $n = 14$; K101R: $n = 16$; V110I: $n = 15$; P148A: $n = 12$; I459V: $n = 11$; L728F: $n = 14$ ($n$, number of plaques). The data are shown as mean ± SD. Statistical significance was determined using two-tailed unpaired $t$ test

($****p < 0.0001$; n.s. $p > 0.05$, not significant). The exact $p$ values are: $p < 0.0001$ (K101R), $p = 0.2885$ (V110I), $p = 0.0526$ (P148A), $p = 0.4413$ (I459V), and $p = 0.8037$ (L728F). **e** Neurovirulence of ZIKV mutants in neonatal mice. CD-1 neonatal mice were i.c. inoculated with either 10 PFU of indicated viruses or PBS, and the mortality was observed for 21 days. WT: $n = 21$; K101R: $n = 12$; V110I: $n = 13$; V125I: $n = 11$; P148A: $n = 11$; I459V: $n = 15$; L728F: $n = 13$; PBS: $n = 10$ ($n$, number of mice). Statistical significance was determined using Log-rank test ($***p < 0.001$; n.s., $p > 0.05$, not significant). The exact $p$ values are: $p = 0.0005$ (K101R), $p = 0.9203$ (V110I), $p = 0.2072$ (P148A), $p = 0.2702$ (I459V), and $p = 0.2665$ (L728F). Source data are provided as a Source Data file.

amino acid substitutions including K101R, V110I, P148A, I459V, I607V, and L728F were selected for further site-directed mutagenesis (Fig. 1b).

**Recovery and characterization of recombinant mutant ZIKVs**
To identify critical amino acid substitutions responsible for the lineage-specific virulence phenotype, mutagenesis analysis was performed using a full-length infectious cDNA clone of a representative Asian lineage strain that contained the S139N and M2634V substitutions[9] as the wild-type (WT) virus. By using reverse genetics, a

total of 6 recombinant ZIKVs, including K101R, V110I, P148A, I459V, I607V, and L728F, were recovered in BHK-21 cells (Fig. 2a). RNA transfection revealed, except for I607V, the other 5 mutant viruses displayed pattern of viral protein expression in Vero cells comparable to the WT virus (Fig. 2b). Plaque forming assay showed that the K101R mutant virus produced significantly larger plaques than the WT virus in BHK-21 cells (Fig. 2c, d). We further tested the mouse neurovirulence of all the recovered mutant viruses by i.c. inoculation of 1-day-old CD-1 neonatal mice with 10 PFU of the mutant viruses. As shown in Fig. 2e,

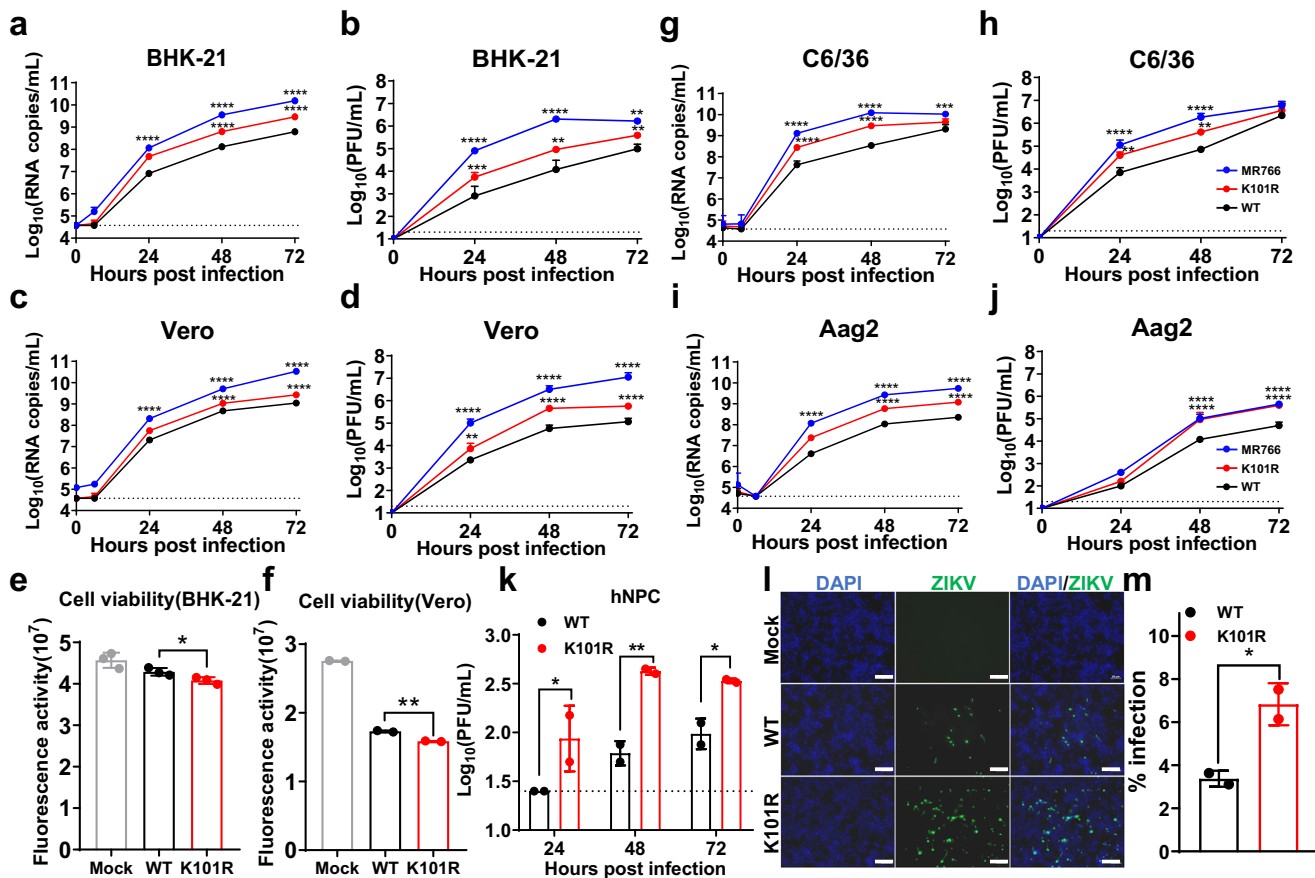

**Fig. 3 | In vitro characterization of the K101R mutant virus. a–d** Growth curves of indicated viruses in BHK-21 and Vero cells. Cells were infected with indicated viruses at an MOI of 0.1. Viral RNA and infectious virus particles in the supernatants at indicated times were measured using RT-qPCR and plaque assay in BHK-21 cells, respectively. The data are shown as mean ± SD. Statistical significance was determined using Two-way ANOVA (**$p < 0.01$, ***$p < 0.001$, ****$p < 0.0001$). The exact $p$ values are (**a**) MR766, $p < 0.0001$; K101R, $p < 0.0001$. **b** MR766, $p < 0.0001$ (24 and 48 h), $p = 0.0052$ (72 h); K101R, $p = 0.0002$ (24 h), $p = 0.0011$ (48 h), $p = 0.0067$ (72 h). **c** MR766, $p < 0.0001$; K101R, $p < 0.0001$. **d** MR766, $p < 0.0001$; K101R, $p = 0.0022$ (24 h), $p < 0.0001$ (48 and 72 h). **e, f** Viability of BHK-21 and Vero cells was detected after infection with WT or K101R at an MOI of 0.01 at 48 h post-infection. The data are shown as mean ± SD. $N = 2$ independent biological replicates (**f**). Statistical significance was determined using two-tailed unpaired $t$ test (*$p < 0.05$, **$p < 0.01$). The exact $p$ values are 0.0416 (**e**) and 0.0059 (**f**). **g–j** Growth curves of indicated viruses in C6/36 and Aag2 cells. Cells were infected with indicated viruses at an MOI of 0.1. Viral RNA and infectious virus particles in the supernatants at indicated times were measured using RT-qPCR and plaque assay in BHK-21 cells, respectively. The data are shown as mean ± SD. Statistical significance

was determined using Two-way ANOVA (*$p < 0.05$, **$p < 0.01$, ***$p < 0.001$, ****$p < 0.0001$). The exact $p$ values are (**g**) MR766, $p < 0.0001$(24 and 48 h), $p = 0.0003$ (72 h); K101R, $p < 0.0001$(24 and 48 h). **h** MR766, $p < 0.0001$ (24 and 48 h); K101R, $p = 0.0032$ (24 h), $p = 0.0028$ (48 h). **i** MR766, $p < 0.0001$; K101R, $p < 0.0001$. **j** MR766, $p < 0.0001$; K101R, $p < 0.0001$. **k** Growth curves of indicated viruses in hNPCs. Cells were infected with indicated viruses at an MOI of 1. Infectious virus particles in the supernatants on indicated times were detected using plaque assay in BHK-21 cells. L.O.D., limit of detection. $N = 2$ independent biological replicates. The data are shown as mean ± SD. Statistical significance was determined using Two-way ANOVA (*$p < 0.05$, **$p < 0.01$). The exact $p$ values are: $p = 0.0463$ (24 h), $p = 0.0059$ (48 h), and $p = 0.0448$ (72 h). **l, m** Representative images (L) from IFA and percentages of infected hNPCs. Cells were infected with indicated viruses at an MOI of 1, the E protein expression of viruses was detected by IFA with ZIKV-E-specific mAbs at 48 h post-infection. The infection rate was determined by counting the ZIKV-E-positive cell foci as a percentage of cells. The data are shown as mean ± SD. Statistical significance was determined using Two-tailed unpaired $t$ test (*$p < 0.05$). The exact $p$ value is 0.0427. Scale bar: 50 μm. Source data are provided as a Source Data file.

mice inoculated with the K101R mutant virus had an average survival time (AST) of 14.2 days, and all mice succumbed to the challenge within 12 to 19 days. In contrast, when compared to K101R, mouse survival for WT was significantly higher with an AST of 17.8 days during the observation period. All the other mutant viruses–inoculated mice displayed a similar pattern of survival to that of WT-inoculated mice (Fig. 2e). Overall, the data indicate only the K101R mutant virus with the larger plaque phenotype exhibits a significantly enhanced mouse neurovirulence in comparison with the WT virus.

## The K101R mutation leads an enhanced replication of ZIKV in cell cultures

To investigate the role of K101R mutation in viral replication in cell culture in detail, growth curves of K101R, WT, and MR766 in multiple

mammalian and mosquito cell lines were performed. As shown in Fig. 3a–d, all viruses replicated efficiently in BHK-21 and Vero cells. However, in terms of both the levels of viral RNA and infectious viral particles, K101R showed significantly higher titers than that of WT, while MR766 displayed the most robust replication. The cell viability assay showed the K101R mutation caused significantly more severe CPE than the WT virus in BHK-21 and Vero cells (Fig. 3e, f). Similar results were observed in mosquito cell lines including C6/36 and Aag2 (Fig. 3g–j). Furthermore, human neural progenitor cells (hNPCs), the main target cells of ZIKV infection[25,26], were used to test the replication of these two viruses in neural cell lines. As expected, the K101R virus replicated more efficiently than the WT virus (Fig. 3k), and immuno-fluorescence assay showed a significantly higher percentage of K101R-positive cells than WT (Fig. 3l, m). These findings support that the

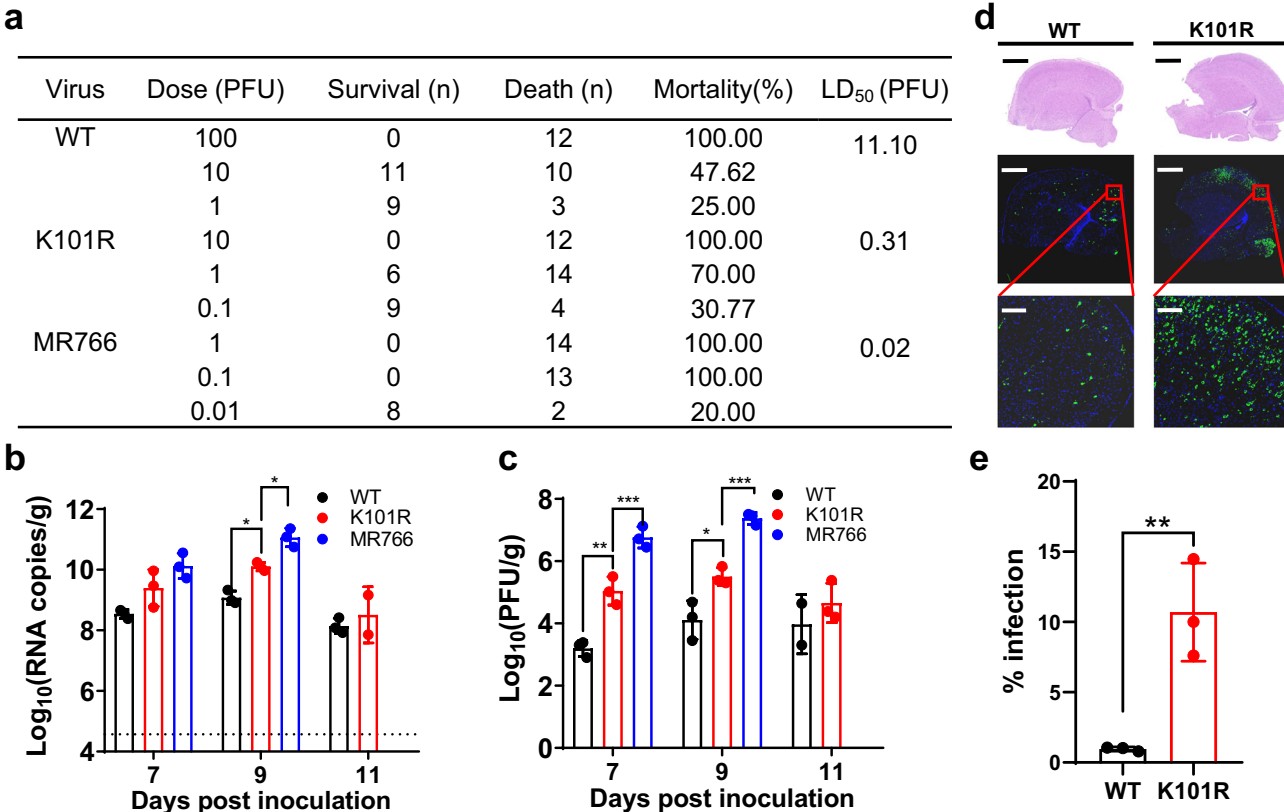

**Fig. 4 | Neurovirulence of the K101R virus in the neonatal mouse model.**
**a** Neurovirulence of viruses in suckling CD-1 P1 mice. One-day-old mice were i.c. inoculated with indicated doses of viruses, The LD$_{50}$ values were calculated by the Reed–Muench method[52]. The pups were monitored for mortality for 21 days.
**b, c** Viral loads in brains of WT-, K101R- or MR766-infected CD-1 mice. Viral RNA and infectious virus particles in mouse brains were quantified by RT-qPCR and plaque assay, respectively. The data are shown as mean ± SD. Statistical significance was determined using Two-tailed unpaired $t$ test (*$p < 0.05$, **$p < 0.01$, ***$p < 0.001$). The exact $p$ values are (**b**) MR766, $p = 0.0346$. K101R, $p = 0.0385$; (**c**) MR766, $p = 0.0004$ (day 7), $p = 0.0002$ (day 9). K101R, $p = 0.0044$ (day 7), $p = 0.0264$ (day 9).

**d** Representative images of coronal plane of brain tissue at day 9 post-inoculation. Top panels show tissues stained with Hematoxylin and Eosin (H&E). Scale bar: 1000 μm. Middle panels show tissues stained with ZIKV E antibody (green) and DAPI (blue). Scale bar: 1000 μm. Down panels represent enlarged areas from red box in the middle of the cortex. Scale bar: 100 μm. **e** The E protein expression of viruses was detected by IFA with ZIKV-E-specific mAbs at 48 h post-infection. The infection rate was determined by counting the ZIKV-E-positive cell foci as a percentage of cells. The data are shown as mean ± SD. Statistical significance was determined using Two-tailed unpaired $t$ test (**$p < 0.01$). The exact $p$ value is 0.0084. Source data are provided as a Source Data file.

K101R substitution in the C protein enhances the replication of ZIKV in multiple cell lines.

## The K101R substitution enhances neurovirulence of ZIKV in neonatal mice

In order to investigate the impact of K101R substitution on neurovirulence, we conducted a detailed examination using a well-established neonatal mouse neurovirulence model[13]. The 50% lethal doses (LD$_{50}$) of the K101R, WT, and MR766 viruses in CD-1 neonatal mice were determined, respectively (Fig. 4a). As expected, MR766 displayed the highest level of neurovirulence, with an LD$_{50}$ of 0.02 PFU. Meanwhile, K101R showed enhanced neurovirulent compared with the WT virus (approximately 35-fold), but still less neurovirulent than MR766 (Fig. 4a). Consistent with this, K101R showed a significantly increased accumulation of viral RNA and infectious particles in mouse brains in comparison with WT, with a difference about 1-2 logs on day 9 post-inoculation (Fig. 4b, c). Additionally, immunofluorescent staining assay using a ZIKV-specific antibody showed significantly increased ZIKV-positive signals in brain sections from the K101R-inoculated mice in comparison with those from WT-inoculated mice (Fig. 4d, e). Together, these data clearly demonstrate that the

K101R substitution in ZIKV C protein contributes to the increased neurovirulence in neonatal mice.

## Transcriptomic analysis of mouse brain infected with the K101R and WT viruses

To further investigate the underlying mechanism accounting for the distinct neurovirulence phenotype, transcriptomic profiles of brains from the K101R- and WT-inoculated mice were performed via RNA-seq analysis and compared with mock infection animals. Differentially expressed genes (DEGs) analysis showed that both viruses induced dramatic changes in gene expression compared with their mock counterparts, and K101R elicited a more robust response. In K101R-infected animals, a total of 2498 and 664 genes were up-regulated and down-regulated, respectively, whereas only 1187 and 109 genes were up-regulated and down-regulated in the WT-infected animals, respectively (Fig. 5a, b). Notably, a large panel of genes related to antiviral innate response, including ifit1, OAS1, ccl5, and ccl10, were up-regulated in both WT- and K101R-inoculated mouse brains. Next, Gene Ontology (GO) analysis of the DEGs was performed to identify potential alterations in signaling pathways and cellular processes upon viral infection. The top 5 upregulated genes in WT-infected brains were

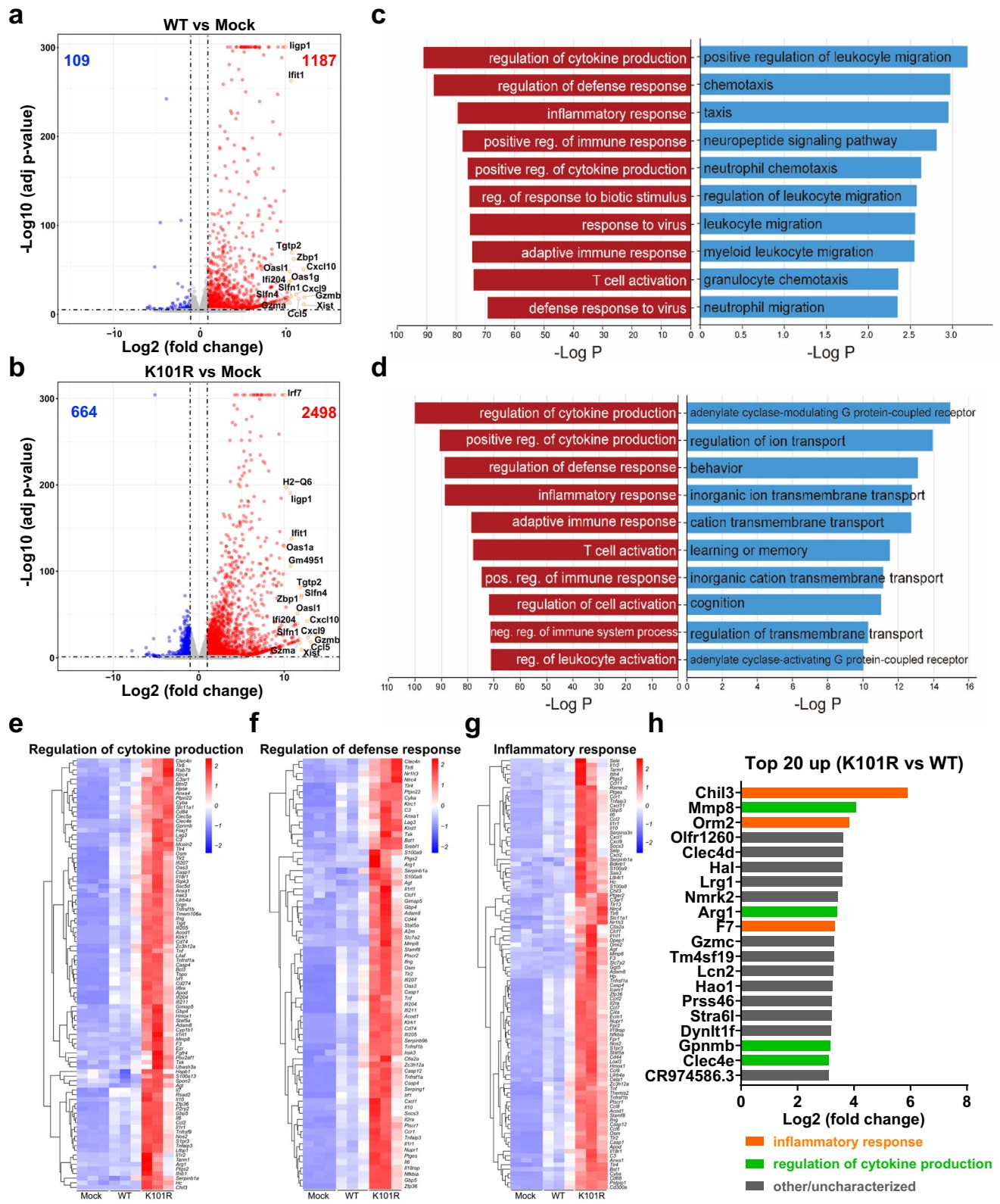

**Fig. 5 | Transcriptomic analysis of brain tissues from ZIKV-infected mice.** CD-1 neonatal mice were inoculated with PBS (Mock), WT or K101R (1 PFU). The samples were harvested on 11 days post-infection. Nine mice were analyzed in total (three for each group). **a**, **b** Volcano plots indicating differentially regulated genes of WT- or K101R- infected mouse brains. The numbers of downregulated and upregulated genes were labeled in blue and red, respectively. The mainly upregulated genes in both figures were labeled with gene symbols. **c** Top 10 Gene Ontology (GO) terms of upregulated and downregulated genes in WT versus mock-infected brains. **d** Top 10 GO terms of upregulated and downregulated genes in K101R versus mock-infected brains. **e**–**g** Heatmap analyses indicated that these upregulated genes were primarily enriched in regulation of cytokine production and defense response as well as inflammatory response. **h** The top 20 upregulated genes in K101R- versus WT-infected brains of mice. Source data are provided as a Source Data file.

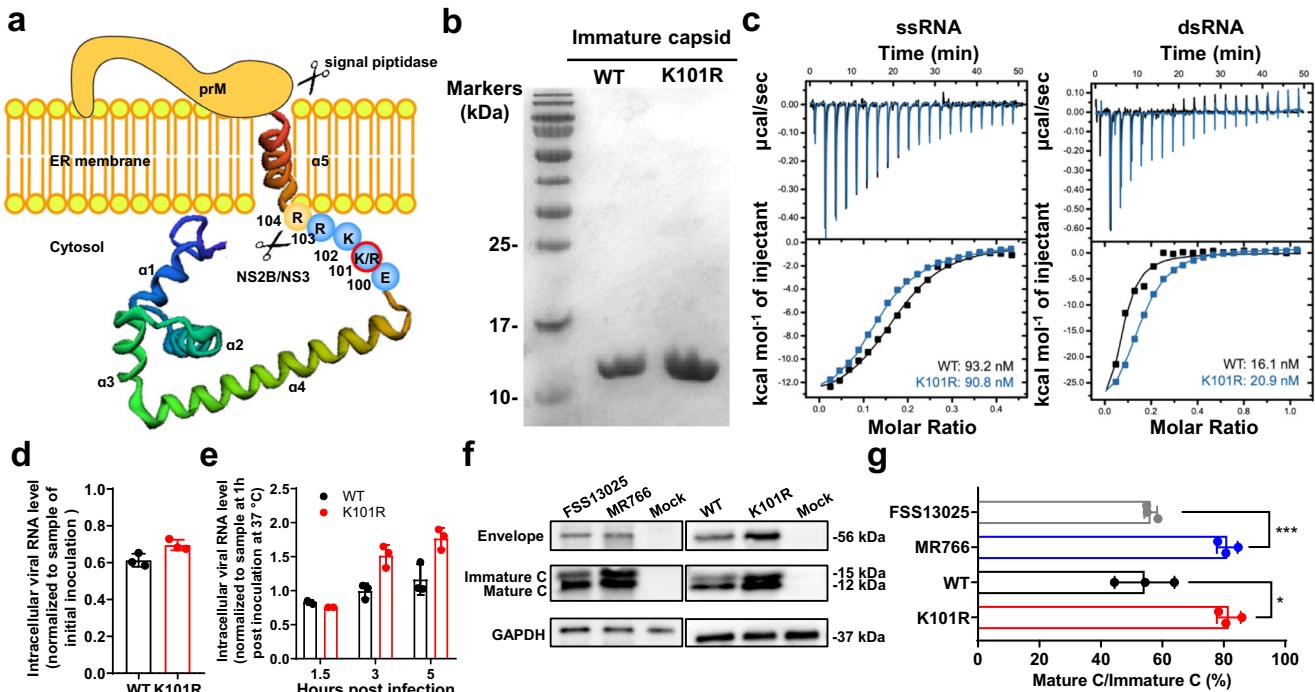

**Fig. 6 | The K101R substitution leads to increased production of mature C protein of ZIKV. a** A model depicting how NS2B/NS3 cuts C-anchor C-prM in the context of 101 K and 101 R constructs. **b** The immature C proteins of WT and K101R were synthesized in vitro and purified through Ni-chelating affinity chromatography. Both C proteins were displayed by SDS-PAGE. **c** Binding properties of C proteins of WT or K101R to ssRNA and dsRNA. The titration data of ZIKV C protein with each of four oligonucleotides were obtained with a MicroITC200 calorimeter at 25 °C. Isothermal titration calorimetry (ITC) assays of the C proteins to ssRNA. **d**, **e** Virus-cell binding and entry. BHK-21 cells were incubated with indicated viruses at an MOI of 1 at 4 °C for 1 h. Following this, virus inocula were removed. Then the cells were washed three times with PBS, replenished with medium, and incubated at 37 °C to allow viruses entry to cells. At 1, 1.5, 3, and 5 h post-incubation, the infected cells were quantified for intracellular viral RNAs using RT-qPCR. Levels of intracellular viral RNA were normalized to the sample at 1 h post-inoculation at 37 °C. **f**, **g** Representative western blotting images of expression of C proteins in BHK-21 cells infected with indicated viruses (**f**) and quantification (**g**). BHK-21 cells were infected with FSS13025, MR766, WT, or K101R at an MOI of 0.1, and the cell lysis solution was collected for at 48 h post-infection. The intracellular levels of E and C proteins were analyzed by Western blotting. Quantification of bands was performed using ImageJ software[50]. The data are shown as mean ± SD. Statistical significance was determined using Two-tailed unpaired *t* test (*$p < 0.05$, ***$p < 0.001$). The exact *p* values are 0.0106 (*) and 0.0004 (***). Source data are provided as a Source Data file.

largely related to regulation of cytokine production, regulation of defense response, inflammatory response, positive regulation of immune response, with additional enrichment of genes involved in positive regulation of cytokine production (Fig. 5c). In brains infected with the K101R virus, the top 5 upregulated genes were mainly enriched in the regulation of cytokine production, positive regulation of cytokine production, regulation of defense response, inflammatory response and adaptive immune response (Fig. 5d). In terms of downregulated genes, the genes in WT-infected brains were largely related to positive regulation of leukocyte migration and chemotaxis (Fig. 5c). In contrast, the downregulated genes in K101R-infected brains were mainly enriched in functions related to G protein-coupled receptor signaling pathway, regulation of ion transport and behavior (Fig. 5d). Compared with those of WT-infected brains, the significantly upregulated DEGs of K101R-infected brains belong to three enriched GO terms, including regulation of cytokine production and defense response as well as inflammatory response (Fig. 5e–g). Among them, the top 20 upregulated genes included Chil3, Mmp8, Orm2, Arg1, F7, Gpnmb, and Clec4e in K101R versus WT-infected mouse brains, which were mainly involved in inflammatory response and regulation of cytokine production (Fig. 5h). Collectively, these findings suggest that both viruses induces robust changes in the mRNA transcriptome in mouse brains, but K101R has a broader and more robust effect on the expression of genes related to innate immune and inflammatory response than WT.

## The K101R substitution increases the production of mature C protein

The Flavivirus C protein is responsible for viral genome RNA packaging. As shown in Fig. 6a, the K101R substitution is located between helix α4−α5 junction of the C protein, where there is the viral protease NS2B/NS3 cleavage site critical for releasing the mature C protein. Furthermore, position 101 immediately follows the helices α4 that interacts with the viral RNA genome. To make sure whether K101R substitution altered the RNA binding capability of C protein, isothermal titration calorimetry (ITC) assay was performed with recombinant C proteins of WT and K101R (Fig. 6b) and viral RNAs from the 5′UTR as previously described[27]. The results showed that both C proteins showed high binding affinities (in the nanomolar range) to ssRNA and dsRNA (Fig. 6c). Furthermore, no significant difference in the virus-cell binding and entry was detected between K101R or WT (Fig. 6d, e), indicating the K101R substitution did not affect virus-cell binding and entry. Lastly, we probed whether the K101R substitution affected the maturation of C protein as previously described[28]. As shown in Fig. 6f, g, Western blotting assay showed a higher ratio of mature C protein to immature C protein in MR766-infected cells than FSS13025 (80% vs 58%), indicating a more efficient protease cleavage in C protein of the African ZIKV. Similarly, the K101R virus produced more mature C protein than the WT virus with a higher ratio of mature C protein/immature C protein (78% vs 52%). Together, the K101R mutation results in higher production of mature C protein without affecting its binding affinity to viral RNA.

## Discussion

The virulence determinants of the more pathogenic African lineage of ZIKV remain to be identified. Here, by using sequence alignment and site-directed mutagenesis, we identify the residue at position 101 of the C protein as a neurovirulent determinant for the African lineage. Mechanistically, we reveal the K101R substitution results in a significantly increased production of the mature C protein without affecting its RNA binding affinity and viral-cell binding and entry.

The main goal of the present study is to examine and determine critical residues for virulence of the African lineage strains. Our research focused on the structural proteins due to their roles in binding, entry, assembling and modulating viral infection cycle[29]. Sequence alignment of representative genomes of ZIKV strains, which are from the African and Asian lineages grouped by phylogenetic analysis, revealed 11 highly conserved residues within the structural proteins. We further found a single K101R substitution greatly increased the neurovirulence of the Asian strain, as evidenced by increased replication in hNPCs and suckling mouse brains, and higher mortality of mice upon i.c. inoculation with the K101R virus. The four other mutant viruses exhibited similar neurovirulence phenotypes to the Asian strain. By swapping the structural and non-structural genes between the African and Asian lineages, Nunes et al provided experimental evidence the function of viral structural genes in determining the virulence difference between the two lineages in mouse models[24]. Nakaya E, et al also reported the higher virulence of the African strain MR766 was associated with prM[30]. Furthermore, a previous study revealed a residue at position 106 of the C protein as a key determinant of ZIKV infectivity in mosquito vectors[31]. Compared with the reports mentioned above, our study further identified a residue within the C protein that is likely to participate in the neurovirulence of the African lineage. Notably, the $LD_{50}$ value of MR766 remains over 15-fold lower than that of K101R (0.02 PFU vs 0.31 PFU), suggesting that additional amino acid substitutions or RNA elements likely coordinately contribute to the neurovirulence phenotype of the African lineage[32,33]. Despite this, to our knowledge, this is the first report on the determination of the key residue of the C protein for neurovirulence of the African lineage.

During viral replication, the anchored C protein (1–122 amino acids) is cleaved by viral protease NS2B/NS3 to release a mature C protein (1–104 amino acids). This mature C protein then assembles with the viral RNA genome in order to facilitate its packaging into the virus particle[34,35]. The residue K101 is located at the P5 position (upstream of NS2B/NS3 cleavage site) according to the Schechter and Berger nomenclature[36], and it is downstream of the helix α4 of the C protein, which is responsible for its direct binding to the genome RNA (Fig. 6a). Therefore, it is reasonable to speculate the mutation at this position is likely to affect the mature process of the C protein or/and the RNA binding ability. Surprisingly, no difference was observed in the RNA binding ability between the K101R and WT, suggesting the electrostatic potentials of helix α4 of the C protein are likely not affected by the K101R substitution. Despite this, a significant difference in the production of the mature C protein was observed between the K101R and the WT, with higher levels of mature C protein in K101R-infected compared to WT-infected cells (Fig. 6b). A previous study has shown that the sequence surrounding critical residues within WNV NS2B/NS3 cleavage site can influence protease specificity[37]. Yu et al also showed that a T106A substitution in the C protein rendered it a preferred substrate for the NS2B-NS3 protease, thereby facilitating the maturation of structural proteins and the formation of infectious viral particles[31]. Based on these findings, it can be postulated that the K101R substitution is likely to alter the dibasic motif at the P1 and P2 positions, potentially affecting the efficiency of C protein cleavage. Both of our study and the report by Yu et al underscore the importance of amino acids immediately next to the NS2B-NS3 protease cleavage site

in the biological functions of ZIKV. However, further investigation is required to understand how the K101R substitution specifically affects the maturation of the C protein. Additionally, while it is unclear whether flavivirus C proteins are directly ubiquitylated, previous studies have shown that ubiquitylation was required for the disassembly of the flavivirus nucleocapsid during the post-fusion step of virus entry[38,39]. Therefore, future work would be needed to investigate whether the K101R substitution influences the ZIKV uncoating via the ubiquitin-proteasome pathway.

Neurovirulence refers to the ability of a viral infection to cause central nervous system (CNS) pathology. The severity of the neurologic symptoms is generally dependent on the virus replication in CNS and/or host immune responses. Mounting evidence suggests the role of the C protein in impairing the innate immune response. For example, the C protein of YFV hinders RNA silencing in the mosquito Aedes aegypti by interfering with Dicer[40]. In addition, ZIKV C protein has been reported to subvert the type I IFN response[41]. Therefore, an enhanced production of mature C protein is likely to suppress antiviral responses in hosts, thereby facilitating more efficient viral replication. On the other hand, the transcriptome profiling in our study revealed that both K101R and WT infections induced robust activation of proteins involved in antiviral immune response (OAS family and IFIT-1), as well as inflammation-related cytokines and chemokines such as CCL5, CXCL9, and CXCL10, when compared to the mock-infected group. Importantly, compared to WT-infected brains, the K101R infection resulted in significantly higher expression levels of genes involved in the regulation of cytokine production and inflammatory response, including Chil3, Arg1, and Mmp8. Chil3 is a secretory protein with chemotactic activity for T lymphocytes, bone marrow polymorphonuclear leukocytes, and eosinophils. The Chil3 and Arg1 have been reported to play a regulatory role in the alternative activation of macrophages during viral infection[42,43]. Recently, Zhang et al reported that brain-infiltrated monocytic myeloid-derived suppressor cells isolated from JEV-infected mice hindered T cell proliferation through Arg1, partly contributing to exacerbated pathogenicity[44]. The higher levels of Chil3 and Arg1 transcripts in K101R-infected mice suggested stronger immune responses during the course of infection compared to WT, partly due to increased replication and enhanced neurovirulence for mice. Furthermore, several studies have implicated the involvement of matrix metallopeptidases (MMPs) in blood-brain barrier (BBB) damage related to neurological disorders. For example, WNV increases the levels of MMPs, leading to enhanced BBB permeability[45]. JEV-infected astrocytes release MMP2/MMP9, which disrupts BBB integrity[46]. MMP8 can modulate BBB permeability change during rabies virus infection[47]. The higher level of Mmp8 transcript in K101R-infected mouse brains likely contributes to more severe disruption of BBB integrity compared to WT, and further work is necessary to ascertain whether BBB permeability changes occur and the underlying mechanisms. Overall, based on the aforementioned data, we propose a model to elucidate the contribution of K101R substitution in C protein to increased ZIKV replication in vitro and enhanced neurovirulence in mice (Fig. 7). In this model, the K101R mutation leads to increased production of matured C protein, which in turn promotes enhanced virus replication in cells and mouse brains.

In conclusion, we identify and characterize a key determinant for the neurovirulence of the ZIKV African strain. This critical residue is located between the helixs α4 and α5 of the C protein. To our knowledge, this is the first report on the single residue of the C protein contributing to the virulence properties of the African lineage. Forthcoming research is required to clarity roles of other highly conserved residues of structural proteins in virulence of the African lineage, thereby offering insights into its biological relevance. These findings will undoubtedly advance our understanding of ZIKV disease pathogenesis, immune responses, and antiviral strategies.

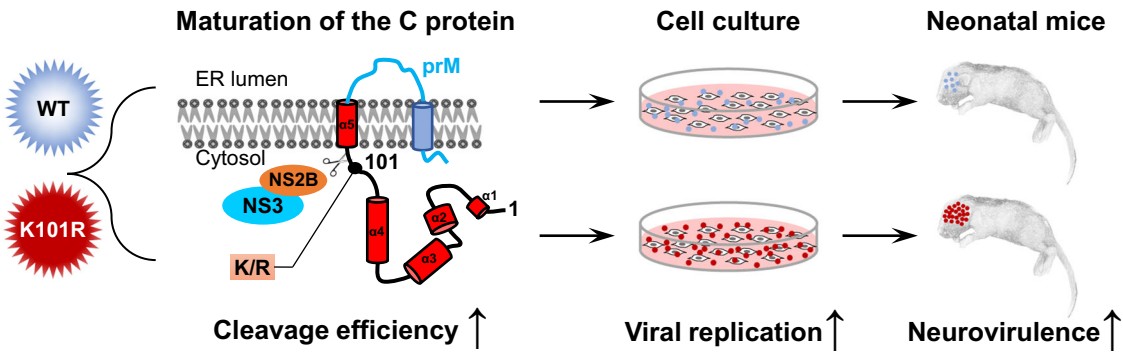

**Fig. 7 | The model for the contribution of K101R substitution in C protein to virulence of ZIKV.** The K101R mutation leads to more production of matured C protein, resulting in the increased virus replication in cells and mouse brains.

## Methods

### Ethics statement
All animal experiments were performed in strict accordance with the guidelines of the Chinese Regulations of Laboratory Animals (Ministry of Science and Technology of People's Republic of China) and Laboratory Animal-Requirements of Environment and Housing Facilities (GB 14925-2010, National Laboratory Animal Standardization Technical Committee). All procedures were approved by the Animal Experiment Committee of Laboratory Animal Center, Academy of Military Medical Sciences, China (IACUC-IME-2021-010).

### Cells and viruses
All cell lines were authenticated by the ATCC. Baby hamster kidney fibroblast cell line BHK-21 (ATCC, CCL-10) and African green monkey kidney cell line Vero (ATCC, CCL-81) were maintained in DMEM (Thermo Fisher Scientific, 2263284) supplemented with 10% fetal bovine serum (FBS, Gibco, 2138109RP) and 1% penicillin/streptomycin at 37 °C with 5% CO$_2$. The *Aedes albopictus* cells C6/36 (CRL-1660) were cultured in RPMI 1640 (Thermo Fisher Scientific, 11875093) medium containing 10% FBS at 30 °C. The *Aedes aegypti* cells Aag2 (ATCC, CCL-125) were cultured in Schneider's Drosophila medium (Thermo Fisher Scientific) containing 10% FBS at 30 °C. Human neural progenitor cell line hNPC 15167 derived from fetal brains (Lonza) was kindly provided by S. Bao (Cleveland Clinic) and cultured as neurospheres in NeuroCult™-XF Basal Medium (STEMCELL, 05760) supplement with basic fibroblast growth factor (bFGF, STEMCELL, 78003.1, 10 ng/mL), epidermal growth factor (EGF, STEMCELL, 78006.1, 20 ng/mL) and Heparin solution (STEMCELL, 07980, 0.1%).

The Asian ZIKV strain FSS13025 (GenBank number KU955593) was isolated from a patient in Cambodia in 2010, and recovered from an infectious cDNA clone[48]. The Africa ZIKV strain MR766 (GenBank number LC002520) was originally isolated from a sentinel monkey in Uganda in 1947, and recovered from an infectious cDNA clone[49]. All viral stocks were prepared in C6/36 cells and titrated by standard plaque-forming assay. All experiments using viruses were performed under biosafety level 2 (BSL-2) conditions at the Beijing Institute of Microbiology and Epidemiology with Institutional Biosafety Committee approval.

### Cell viability
Utilizing the CellTiter-Glo® Cell Viability Assay (Promega), the activity of ZIKV was tested. Cells in a 24-well plate at a density of $1.5 \times 10^5$ cell/well were infected with WT, K101R, FSS13025, or MR766 strains respectively at an MOI of 0.01, while DMEM media was utilized for Mock cells. Following 48 h of incubation in a 37 °C, 5% CO$_2$ incubator, 200 μL of CellTiter Glo® reagent was added to each well. Then the fluorescence (Luminescence) value was recorded to gauge cell viability.

### Plaque assay
Virus samples were serial-diluted by 10-fold with DMEM containing 2% FBS. Then 400 μL of each dilutes were added onto BHK-21 cell monolayers in 12-well plates and incubated at 37 °C under 5% CO$_2$ for 1 h. The supernatants containing the viruses were discarded and 1 mL of DMEM containing 1% low melting point agarose (Promega, V2111) and 2% FBS was added into each well. The infected cells were cultured for another 4 days and then fixed with 4% paraformaldehyde (PFA, Biosharp, BL539A), followed by staining with 1% crystal violet solution. Plaques were counted for the calculation of virus titers.

### Phylogenetic analysis of ZIKV genomes
The approach utilized for the filtration of all ZIKV genomes available on the GenBank was as follows: Initially, a comprehensive collection of genomes was assembled, focusing on those associated with the taxonomic identifier taxid:64320. This primary collection amounted to a total of 2193 genomes (accessed on 2023-07-03). Subsequently, only those genomes falling within a specific length range of 10200 to 10900 base pairs and containing fewer than 10 ambiguous nucleotide characters (represented by the symbol 'N'), were chosen for further analysis. Genomes linked to patents, vaccines, or labeled as 'UNVERIFIED' were selectively omitted from the study. Following these rigorous quality control measures, a total of 722 genomes were found to meet the specified criteria and were considered suitable for subsequent analysis. Out of these genomes, 685 were identified as originating from the Asian lineage, while the remaining 37 were linked to African lineages. To select representative genomes of the Asian lineage, the CD-hit program (v4.8.1) was utilized, with a sequence identity threshold of 0.995. This procedure resulted in the identification of 43 representative genomes from the pool of 685 Asian lineage strains. Manual curation was conducted for the African lineage, where representative genomes were hand-picked for each strain, with emphasis placed on selecting genomes with the least complex passage history. Consequently, a final selection of 52 genomes, consisting of 9 African lineage and 43 Asian lineage genomes, were cautiously chosen as representative genomes and subsequently subjected to further analysis. The maximum likelihood phylogenetic tree of 52 ZIKV complete sequences was constructed using IQ-Tree (v1.6.12) with 1000 ultrafast bootstrap tests. GTR + F + G4 substitution model was chosen as the Best-fit model according to Bayesian Information Criterion (BIC). Bootstrap values were assigned to tree nodes if they exceeded 80.

### Generation of ZIKV mutants
the Q5® site-directed mutagenesis kit (NEB, E0552S) was utilized to introduce single amino acid substitutions (K101R, V110I, P148A, I459V, I607V, and L728F) into the infectious clone of FSS13025 with S139N and M2634V mutants (WT). The infectious clone plasmids were linearized by restriction endonuclease digestion and purified by Phenol/

Chloroform extraction. In vitro transcribed viral RNA was prepared using Ribomax T7 large-scale RNA production kit (Promega, P1300) and purified using Purelink RNA mini kit (Thermo Fisher Scientific, 12183018 A). The RNA was then transfected into BHK-21 cells using Lipofectamine 3000 reagent (Thermo Fisher Scientific, L3000001). Culture supernatants were collected after 48–72 h post-transfection, and infectious virions were detected by plaque assay and viral antigen expression was detected with indirect immunofluorescent assay. The titers of virus stocks were determined by plaque assay, and the substitution sites were confirmed by RT-PCR (PrimeScript™ One Step RT-PCR Kit, TaKaRa, RR055A) and DNA sequencing.

### Manual plaque diameter measurements

Photographs of plates displaying plaques of various sizes (one plate per image) were captured on a viewbox employing a camera equipped with a standard filter, white epi-illumination options, and an image exposure time of 1/160s. Plaques were manually identified and quantified using ImageJ version 1.51j8, utilizing the 'Analyze/Measure/Oval' feature. Prior to measurement, each image was magnified up to 300%, and the plaques were manually delineated using the 'Oval' feature. The plaques were then assigned identifiers matching those on the PST output image using the 'Analyze/Tools/ROI Manager'. Once all plaques were selected, their areas were measured using the Region of Interest (ROI) Manager. Finally, the diameters were calculated based on their respective areas.

### Growth curves

BHK-21, Vero, C6/36, Aag2 cells, and hNPCs were seeded onto 24-well plates one day before. The culture supernatants were replaced with virus-containing medium at an MOI of 0.1, except for the hNPCs which were infected at an MOI of 1. After an incubation at 37 °C for 1 h, the viral supernatants were removed and 0.5 mL of fresh medium was added to each well (DMEM containing 2% FBS for BHK-21 and Vero; 10 ng/mL EGF (R&D), and 10 ng/mL bFGF (R&D) was used for hNPCs). Infected cells were cultured at 37 °C with 5% $CO_2$ and culture supernatants were collected at the indicated time points. Virus titers were determined by plaque assay. Alternatively, viral RNA was extracted using QIAamp® Viral RNA Mini Kit (QIAGEN, 52906) and a previously established RT-qPCR was performed. Standard curves were generated using a 10-fold serial dilution of In vitro transcribed ZIKV viral RNA for the quantification of copies of the ZIKV genome present in samples.

### ZIKV RNA quantification by RT-qPCR

Cells were lysed with RNA lysis buffer, and RNA was extracted using the QIAamp® Viral RNA Mini Kit (QIAGEN, 52906) according to the manufacturer's instructions. RNA quantification in each sample was performed by targeting the NS5 gene of ZIKV. ZIKV RNA was quantified by a One-Step PrimeScript™ RT–PCR Kit (Takara, RR064A) with the following primers and probes: ZIKV-NS5-ASF (5′-GGTCAGCGTC CTCTCTCTAATAAACG-3′); ZIKV-NS5-ASR (5′-GCACCCTAGTGTCCACT TTTTCC-3′); and ZIKV-NS5-Probe (5′-AGCCATGACCGACACCA CCCGT-3′).

### Animal experiments

For the mouse neurovirulence test, CD-1 neonatal mice were intracranial injected with either 30 μL of ZIKV or PBS, with inoculation doses specified in figure legends, and subsequently monitored daily for survival status for a period of 21 days. Survival analysis was performed using the Log-rank analysis. For the determination of viral RNA or infectious virus particles, the brain tissues were ground with 0.8 mL of DMEM medium containing 2% FBS, after centrifugation of 12000 rpm at 4 °C for 5 min, the supernatants were collected and stored at -80 °C for later use.

For the preparation of the samples to the immunohistochemistry (IHC) and RNA-seq, CD-1 neonatal mice were intracranial injected with either 30 μL of ZIKV or PBS, with inoculation doses specified in figure legends. Mouse brains were collected at the indicated time points. Some of the brain tissues were fixed with 4% paraformaldehyde (PFA) for the paraffin section, the others were ground with 0.8 mL Trizol (Invitrogen, Carlsbad, CA, USA) and RNA was extracted for RNA-Seq.

### RNA-Seq and data analysis

The RNA library construction and high-throughput sequencing were performed by Beijing Annoroad Gene Technology Company. Briefly, multiplexed libraries were sequenced for 150 bp at both ends using an Illumina HiSeq6000 platform. Clean reads were aligned to the mouse genome (Mus_musculus.GRCm38.99) using Hisat2 v2.1.0. The number of reads mapped to each gene in each sample was counted by HTSeq v0.6.0 and TPM (Transcripts Per Kilobase of exon model per Million mapped reads) was then calculated to estimate the expression level of genes in each sample. Genes with Padj <0.05 and |Log2FC| ≥ 1 were identified as DEGs. DEGs were used as a query to search for enriched biological processes (Gene ontology BP) using Metascape. Heatmaps of gene expression levels were constructed using pheatmap (version 1.0.12) package in R. Dot plots, volcano plots and bar plots were constructed using ggplot2 (https://ggplot2.tidyverse.org/) package in R.

### Immunofluorescence assay

BHK-21 cells grown on the cover slips were fixed in acetone/methanol (v/v: 3/7) at −20 °C. The cover slips were incubated with the primary antibody at 37 °C for 1 h. After the primary antibody incubation, the cover slips were washed with PBS for three times. Then Alexa Fluor 488-labeled goat anti-mouse IgG antibody (1:200 diluted, GeneTex, GTX213111-04) was added and incubated for 1 h, the cover slips were washed as described above. For cell nuclei staining, DAPI (0.5 ng/μL) was added onto the cover slips, which were incubated for 5 min. An Olympus BX51 microscope equipped with a DP72 camera was utilized for image capture. The antibody used in immunostaining was anti-Zika ENV mAb (1:2000 dilution, BioFront Technologies, BF-1176-56).

### Tissue immunofluorescent staining

The 4-μm-thick paraffin sections were deparaffinized in xylene and rehydrated in a series of graded alcohols. Antigen retrieval was performed in citrate buffer (pH = 6) by heating in a microwave (Sharp, R-331ZX) for 20 min at 95 °C followed by a 20 min cool-down period at room temperature. Multiplex fluorescence labeling was performed using TSA-dendron-fluorophores (NEON 9-color Allround Discovery Kit for FFPE, Histova Biotechnology, NEFP950). Briefly, endogenous peroxidase was quenched in 3% $H_2O_2$ for 20 min, followed by treatment with blocking reagent for 30 min at room temperature. The primary antibody was incubated for 2–4 h in a humidified chamber at 37 °C, followed by detection using the HRP-conjugated secondary antibody and TSA-dendron-fluorophores. Then, the primary and secondary antibodies were thoroughly eliminated by heating the slides in retrieval/elution buffer (Abcracker®, Histova Biotechnology, ABCFR5L) for 10 s at 95 °C using a microwave. In a serial fashion, each antigen was labeled with distinct fluorophores. The antibody used in immunostaining was anti-Zika ENV mAb (1:2000 dilution, BioFront Technologies, BF-1176-56).

### Western blotting analysis

The samples were separated using a 15% SDS-PAGE gel, followed by electrophoretic transfer to polyvinylidene fluoride membranes, which were then blocked and incubated with primary antibodies. The following antibodies were used in the experiments: anti-GAPDH antibodies (MA5-15738, Invitrogen, 1:3,000 dilution); anti-C antibodies (GTX133317, GeneTex, 1:2,000 dilution); anti-E antibodies (BF-1176-56, bioFrontTech, 1:2,000 dilution). Quantification of bands was performed using ImageJ software[50].

## Isothermal titration calorimetry (ITC)

All oligonucleotide primers (12 mer ssDNA 5′-AGTTGTTGATCT-3′, 12 bp dsDNA 5′-AGTTGTTGATCT-3′, 12 mer ssRNA 5′-AGUUGUU-GAUCU-3′, 12 bp dsRNA 5′-AGUUGUUGAUCU-3′) were synthesized by Sangon Biotech. Micro-calorimetric titrations were performed at atmospheric pressure at 25 °C in a MicroITC200 (Malvern Instrument Ltd., U.K.). The experiments were performed in an assay buffer composing of 10 mM HEPES, 150 mM NaCl, pH 7.4. To minimize the heat effect of dilution during the injection, both the oligonucleotide primers and C protein solutions were prepared in this buffer. The sample cell (300 μL) was filled with degassed oligonucleotide primer (15 μM) buffer, and the C protein (40 μM) solution (60 μL) was injected from the syringe into the sample cell under the agitation of 600 rpm. The time interval of each injection (2 μL) was 150 s. All data were collected and analyzed using Origin iTC200 software (Malvern Instrument Ltd.).

## Virus-cell binding and entry assay

Cell binding and entry assay were performed as described previously[51]. Briefly, BHK-21 cells were incubated with WT or K101R at an MOI of 1 at 4 °C for 1 h. Following this, virus inocula was removed. Then the cells were washed three times with PBS, replenished with medium, and incubated at 37 °C to allow viruses entry to cells. At 1, 1.5, 3, and 5 h post incubation, the infected cells were quantified for intracellular viral RNAs using RT-qPCR.

## Statistical analysis

Data analysis was carried out using the GraphPad Prism software. Log-rank tests were performed for the survival analysis. For the statistical analysis of other results, statistical evaluation was performed by two-way ANOVA for double factor analysis or Student's unpaired $t$ test for single factor analysis. Data are presented as means ± SD or as described in figure legends. $p$ values are denoted as follows (ns: not significant, $*p < 0.05$, $**p < 0.01$, $***p < 0.001$, $****p < 0.0001$). Exact $p$ values are provided in figure legends. All experiments were repeated at least three times unless otherwise stated. $N = 3$ independent biological replicates were used for all experiments unless otherwise indicated. No data was excluded from the analyses. The Investigators were not blinded to allocation during experiments and outcome assessment.

## Reporting summary

Further information on research design is available in the Nature Portfolio Reporting Summary linked to this article.

# Data availability

All data are available within the article, Supplementary Information or Source Data file. Source data are provided with this paper. Raw RNA-seq data related mouse brains in this study have been deposited in the NCBI Gene Expression Omnibus (GEO) data sets under accession numbers GSE226495. Source data are provided with this paper.

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

## Acknowledgements

This study was supported by the National Key Research and Develop-ment Project of China (2022YFC2604101 (X.F.L.), 2021YFC2300202 (X.F.L.)), the National Natural Science Foundation of China (82171820 (X.F.L.)) and Natural Science Foundation of Jiangsu Province (BK20220278 (H.Y.Z.)). C.F.Q. was supported by the National Science Fund for Distinguished Young Scholar (81925025) and the Innovative Research Group (81621005) from the National Natural Science Foun-dation China.

## Author contributions

C.F.Q. and X.F.L. conceived the study. G.Y.S., X.Y.H., M.J.H., H.Y.Z., R.T.L., Y.T., Y.W., M.L.C., X.C., R.R.Z., C.Z., and J.Z. conducted the experiments. M.J.H. and R.T.L. performed transcriptome profiles analy-sis. G.Y.S., X.Y.H., H.Y.Z. M.J.H., X.Y.F., X.F.L., and C.F.Q. did the data analysis and interpretation. X.F.L. and C.F.Q. wrote the manuscript. All authors reviewed and approved the manuscript.

## Competing interests

The authors declare no competing interests.
