## [Peer Review File · Nature Communications]

A single amino acid substitution in the capsid protein of Zika virus contributes to a neurovirulent phenotypeREVIEWER COMMENTS

Reviewer #1 (Remarks to the Author):

The manuscript "A single amino acid substitution in the capsid protein of Zika virus contributes to the African lineage-specific virulence phenotypes" by Qin et al. describes the identification and characterization of a single a.a. residue involved in increase virulence of ZIKV. Overall the manuscript is well written with only minor editing for the English language required. In addition to the identification, the authors also come up with a possible mechanism. The data support the conclusions.

One major comment, is that to my knowledge, there is no indication that in the field, the African lineage is more severe or more neurovirulent. This could be due to the limited surveillance in areas where this lineage is endemic, however, it does not support the overall claims by the authors. Regardless, scientifically, this is a very interesting finding, and one showing once again that even a single aa change can have a large impact on the outcome.

Major comments:

The African prototype ZIKV strain MR766 has been passaged in mouse brain multiple times and as such has likely adapted to enhanced neurovirulence etc. Therefore, the MR766 strain is not representative for the African lineage.

While there is experimental data suggesting increased virulence of African lineage ZIKV, there is no data suggesting this is the case in the field.

In addition to comparing the K101R mutant with the WT, it should also be compared to MR766 in order to show how much this single mutation recapitulates the observed phenotype of MR766.

Minor comments:

Line 96: misspelling of prototype

Line 101 (Figure 1C): No data on mobility is presented, so why mention? Why was a mock infected control group not included for weight/survival comparison?

Line 104. The observation in MR766 specific, not lineage specific.

Line 106-108. In the materials and methods it states that sequence alignment was done on a subset of sequences, not all available ZIKV genomes. Please specify here. It is also not clear if these were the only 6 residues, or a selection?

Line 138. While significantly higher, it is a less than 1 log difference. Is this really a biologically relevant difference?

Line 138-139 (Figure 3): If this graph is representative from a single experiment how did you carry out reliable statistics to find significance from less than 1 log difference? Why are all three experiments not combined? Why are the titres shown here as bar graphs rather than the growth curves as used in figure 1A?

Line 140. Please include cell viability for the other cells as well, not just vero.

Line 144 (Figure 3D): You have not stated how many experiments were carried out to obtain the data for this graph. In addition, titres are very low – $\sim 1 \times 10^2$ PFU/ML is this expected with cells that should be highly susceptible to ZIKV infection?

Line 145 (Figure 3F): I would not label this axis "positive rate" but "% of infection" or something similar.

Line 147. Instead of enhancing replication, it could also enhance binding or entry, please show that this is not altered.

Line 158. Please include data on infectious virus titers.

Line 159. Please explain why this difference is only observed on day 9?

Line 161: What do the brains of day 11 mice look like?

Line 166-170. What is the rationale for performing the transcriptomics on day 11, while the difference in replication is observed on day 9?

Line 166-198. Please include analysis of unique and common genes. The genes unique to either virus may identify virus specific responses.

Line 206: I don't really understand this rationale – why did you assume that K101R would abolish RNA binding capability? Wouldn't this hamper RNA packaging therefore making the virus less virulent, which is not what you claim with the rest of your data?

Line 216 (figure 6D): You showed increased replication of K101R in the BHK-21 cells, and here it looks like there is indeed more virus present as shown by the intensity of the envelope. Could the ratio be misrepresented by there simply being a higher input of virus? Why not do western blot on a standardized virus input? Also, the ratio of immature to mature looks quite different between FSS13025 vs WT, which is indicated in panel E. Why does this difference exist? Additionally, please correct "Envolope" to "Envelope"

Line 233. In the main text you state "all" ZIKV strains and not selected ZIKV strains.

Line 246: Does the residue at position 106 have a similar function as you have observed with your C protein mutation?

Line 247. What is the effect of the K101R mutation in mosquitoes?

Line 263. How does the increase in production of mature C protein result in increased replication? This suggests that the availability of mature C is the limiting factor in replication?

Line 275: "Might do not change" does not make sense.

Line 282. Same question as line 263. Does neurovirulence depend on replication or host response, or both?

Line 345. How was this analysis performed?

Line 348. Why were these 2 additional mutations included?

Line 399. What volume was used for inoculation? And please include a mock control for effect of inoculation.

Reviewer #2 (Remarks to the Author):

In their manuscript "A single amino acid substitution in the capsid protein of Zika virus contributes to the African lineage-specific virulence phenotypes," Song et al. identify a specific mutation (K101R) to explain the virulence of African lineage Zika virus. Through phylogenetic and sequence analysis, they find distinguishing mutations that could contribute to the virulence. They generate point mutations in Zika infectious clones and find the K101R in C protein increases viral PFU in cell lines and RNA in neonatal mice brains. The K101R has increased C protein cleavage of the K101R mutant compared to WT, which could be facilitating the maturation of structural proteins and formation of infectious viral particles. This is an interesting study but focuses exclusively on African-lineage neurovirulence; it is unclear how it relates to neurovirulence of i.e. the Puerto Rico strain that does not contain the mutation.

Specific comments:

Please include MR766 in the infections presented in Figure 2 and 3 in order to fully compare the virulence of the K101R to both WT and MR766.

Please quantify the percent of cells infected with the different Zika viruses and the GFP MFI. It appears that there are a lot more infected cells (and could be brighter GFP signal) in the K101R compared to WT.

Are there differences in the temperature differences or amount of viral RNA in the mice infected in Figure 2E?

Please provide viral RNA quantification of the infected cells in Figure 3.

Is the mouse experiment in Figure 4 independent of the one presented in Figure 2? Is there survival/temperature data for Figure 4? What about PFUs in the brain?

Please include statistics for Figure 6D/E.

Please discuss the possibility of the effects of the K101R mutation on affecting posttranslational modifications of the C protein.

The Puerto Rico (DB-ZIKV966) lineage from 2016 (Figure 1) is known for its increased virulence and neuropathogenesis, but does not have the K101R mutation. Please discuss how the K101R model fits with the virulence of the PR virus.

Reviewer #3 (Remarks to the Author):

In this article, the authors investigated the determinant(s) of increased virulence of the historical isolate of ZIKV strain of African lineage (MR766) as compared to a contemporary isolate (FSS13025) of Asian lineage. Phylogenetic analysis revealed six African lineage-specific residues in the structural proteins that were different from the Asian lineage virus. Using an infectious clone of FSS13022, individual mutations were introduced into the clone and viruses were characterized for their growth in vitro. Of all the mutants examined, the K101R mutant virus (having an arginine in place of lysine in the C protein) was shown to possess enhanced replication phenotype and exhibited increase neurovirulence properties compared to the wild-type (WT) infectious-clone derived FSS13022 virus. The authors performed RNA-seq analysis of the brains of animals infected with the mutant and WT virus to determine the genes that are up-regulated or down-regulated in response to virus infections. Further, the authors demonstrated that K101R mutation leads to increased production of the mature C protein. Overall, the authors conclude that K101R substitution in C protein is a key determinant of lineage-specific virulence of ZIKV.

Although the results are interesting, the studies are very limited in nature. The paper lacks any demonstration of conceptual advance in our understanding of ZIKV replication, virulence, and pathogenesis. Furthermore, the studies are neither novel nor impactful. I believe the paper does not meet the quality and novelty that are expected of papers published in Nature Communication.

Specific Comments:

1. The title of the paper is misleading as it conveys that only a single mutation in the C protein is responsible for the higher virulence phenotype of African lineage virus. This is certainly not the case.

Many studies in the literature have identified other mutations and proteins that contribute the virulence.

2. Figure 1 presents studies that only reconfirm the existing data in the literature. No novel or new data are presented here. Also, the sequence alignment and phylogenetic analysis used sequences of only about 30 or so (out of hundreds available in database) isolates.

3. Figures 1C and 2E: It is not clear why the authors used two different doses of the virus for i.c. inoculation for neurovirulence studies, particularly when just 1 PFU of MR766 virus was sufficient to kill all mice.

4. Figure 3: The growth differences of WT and K101R viruses in most cases appear to be marginal (although statistically significant).

5. Figure 5: The authors conducted transcriptomic analysis to "... further investigate the underlying mechanism..." (line 167) but following the analysis, the results were presented in a very superficial manner. It is difficult to determine what is specifically up-regulated and what is down-regulated in K101R virus as compared to WT virus. How the differentially expressed genes affect the virulence phenotypes of the two viruses are not discussed. Since the authors used brains of animals infected with the two viruses for the analysis, it would have been interesting to examine genes that are involved specifically in neurogenesis, and/or those that are involved in microcephaly. Overall, these studies and the results, as described here, appear to be simply a fishing expedition with no real hypothesis and clear cut analysis of the data.

6. Figure 6: To their credit, the authors have demonstrated that the K101R mutation results in production of higher levels of mature C protein. But it is not clear how this occurs as both of these residues are positively charged. The authors postulate that "...K101R substitution is likely to change the dibasic motif at the P1 and P2 position..." (line 271). Is it possible that since R is more positively charged as compared to K, the cleavage efficiency is enhanced?

My recommendation: Reject

Responses to the reviewers' comments

Reviewer #1 (Remarks to the Author):

The manuscript “A single amino acid substitution in the capsid protein of Zika virus contributes to the African lineage-specific virulence phenotypes” by Qin et al. describes the identification and characterization of a single a.a. residue involved in increase virulence of ZIKV. Overall, the manuscript is well written with only minor editing for the English language required. In addition to the identification, the authors also come up with a possible mechanism. The data support the conclusions.

Reply: Thanks for the positive comments.

One major comment, is that to my knowledge, there is no indication that in the field, the African lineage is more severe or more neurovirulent. This could be due to the limited surveillance in areas where this lineage is endemic, however, it does not support the overall claims by the authors. Regardless, scientifically, this is a very interesting finding, and one showing once again that even a single aa change can have a large impact on the outcome.

Reply: We totally agree with the reviewer that the clinical evidence supporting a more virulence African lineage remains lacking. Our study was basically designed to answer the most fundamental scientific question why African ZIKV showed enhanced virulence phenotype in most experimental system including cell culture and animal models. Accumulating data has suggested African strains replicated to high titers and displayed more severe or more neurovirulent in multiple mice models than Asian strains ([1, 2] *refs 14-19 in the revised manuscript*). Additionally, several studies showed that African isolates were able to infect the placenta and cause fetal harm in pregnant non-human primates similar to that of Asian-lineage viruses [3-5]. Whether such animal data can directly translate into clinical findings deserves further investigation. Considering the recent reports of Africa ZIKV in new areas, such surveillance and investigation will be of high importance and emergency.

Major comments:

The African prototype ZIKV strain MR766 has been passaged in mouse brain multiple times and as such has likely adapted to enhanced neurovirulence etc. Therefore, the MR766 strain is not representative for the African lineage.

Reply: Thanks for the comment. We agree with the reviewer's concern about the passage history of the MR766 strain. Through multiple passages in mouse brains, it is inevitable that adaptive nucleotide mutations have been introduced into the genome of MR766. Despite this, the MR766 strain is still phylogenetically identified as the African

lineage based on the genome sequence information, together with other low-passage-number isolates (such as Dakar-1984). Furthermore, the MR766 strain still retains multiple highly conserved amino acids that are characteristic of the African lineage [6]. In addition, a growing body of evidence points to a stronger virulence for African strains including the MR766 strain compared to Asian strains [7]. The MR766 used in our study was synthesized and recovered via reverse genetics, and the full genome sequence was confirmed by DNA sequencing. Therefore, the majority of past papers in the field used the MR766 strain as the prototype strain for African ZIKV.

While there is experimental data suggesting increased virulence of African lineage ZIKV, there is no data suggesting this is the case in the field.

Reply: Thanks for the comment. As responded above, our intention was not to draw a direct correlation between our findings and human diseases. The present study was conducted based on established research platforms including *in vitro* cultures and mouse models, and the enhanced virulence phenotype of African lineage has also been validated by other groups [1, 8, 9]. While direct translation of these findings to human disease is not possible, the biological mechanism uncovered in the present study is critical and warrants careful consideration. We have mentioned this point in the revised manuscript (**lines 75-78**)

In addition to comparing the K101R mutant with the WT, it should also be compared to MR766 in order to show how much this single mutation recapitulates the observed phenotype of MR766.

Reply: Thanks for the insightful suggestion. We have performed additional experiments to compare the neurovirulence of MR766 and K101R in suckling mice. The LD₅₀ values of MR766 and K101R for suckling mice were calculated to be 0.024 PFU and 0.31 PFU, respectively. This result clearly demonstrates that the K101R mutation is not the sole virulence determinant of MR766, and some additional amino acid substitutions or RNA elements may also contribute to the neurovirulence phenotype. We also mentioned this point in the Discussion section (**lines 272-275**). In addition, we performed growth curve analysis of viruses including WT, K101R and MR766 in BHK-21 and Vero cells. Viral RNA and infectious viral particles were quantified using RT-qPCR and plaque assay, respectively. K101R exhibited an intermediate replication efficiency between WT and MR766. All these new data were combined in the new **Fig. 3A-3D**, and **Fig. 4A-4C** in the revised manuscript.

Minor comments:

1. Line 96: misspelling of prototype

Reply: Corrected.

2. Line 101 (Fig. 1C): No data on mobility is presented, so why mention? Why was a mock infected control group not included for weight/survival comparison?

Reply: The statement about mobility is based on daily observation. The PBS-inoculated group was also included for comparison. **Fig. S1C** was updated accordingly in the revised manuscript.

3. Line 104. The observation in MR766 specific, not lineage specific.

Reply: Thanks for the comment. The African prototype strain MR766 has been widely used to compare the phenotypic divergence between two distinct lineages [8, 9]. Whatever, we have re-written this sentence as follows ‘Due to the African prototype strain MR766 has been widely used to compare the divergence between two distinct lineages, to identify the potential functional residues responsible for this lineage specific phenotype’ in the revised manuscript (**lines 110-112**).

4. Line 106-108. In the materials and methods it states that sequence alignment was done on a subset of sequences, not all available ZIKV genomes. Please specify here. It is also not clear of these were the only 6 residues, or a selection?

Reply: Thanks for the careful checking. We revised the sentences to specify that “we further performed phylogenetic analysis and sequence alignment of the representative genome sequences of ZIKV strains available in GenBank.” (**lines 112-114**). Sequence alignment revealed 11 lineage-specific residues in the structural proteins of ZIKV, out of which six residues, specifically K101R, V110I, P148A, I459V, I607V and L728F were selected for further mutagenesis. We also updated accordingly the description of selection of representative genomes for Asian and African lineages in “materials and methods” section in the revised manuscript.

5. Line 138. While significantly higher, it is a less than 1 log difference. Is this really a biologically relevant difference?

Reply: Thanks for the comment. We performed additional experiments to further tested the replication of viruses including K101R, WT and MR766 in mammalian and mosquito cell lines as well as in mouse brains. Growth curve analysis in cells showed K101R replicated more efficiently than WT in terms of viral RNAs and infectious viral particles, despite of a less than 1 log difference. Importantly, K101R showed a significantly increased accumulation of viral RNA and infectious particles in mouse brains in comparison with WT, with about 1-2 log difference (new **Fig. 3B and 3C** in the revised manuscript). Consistent with this, the LD₅₀ of K101R for suckling mice showed a reduction of about 350-fold in comparison with that of WT (new **Fig. 3A** in the revised manuscript). Therefore, we conclude that the increased replication of K101R in cells have biological relevance, even with a difference of less than 1 log.

6. Line 138-139 (Fig. 3): If this graph is representative from a single experiment how did you carry out reliable statistics to find significance from less than 1 log difference? Why are all three experiments not combined? Why are the titres shown here as bar graphs rather than the growth curves as used in Fig. 1A?

Reply: Thanks for the comment. To assess the replication efficiency of viruses *in vitro*, we further tested viral RNA and infectious viral particles in the supernatants of BHK-21 or Vero cells infected with WT, K101R or MR766 using RT-qPCR and plaque assays, respectively. The new data were representative of three independent experiments, and were shown in new **Fig. 3A-3C** in the revised manuscript. To ensure consistency of data presentation, the data on viral replication in cells were shown as growth curves as used in new **Fig. S1**.

7. Line 140. Please include cell viability for the other cells as well, not just vero.

Reply: Thanks for the suggestion. The data on the cell viability for BHK-21 were provided as new **Fig. 3I** in the revised manuscript.

8. Line 144 (Fig. 3D): You have not stated how many experiments were carried out to obtain the data for this graph. In addition, titres are very low – $\sim 1 \times 10^2$ PFU/ML is this expected with cells that should be highly susceptible to ZIKV infection?

Reply: The data on replication of viruses including WT and K101R in hNPC cells are representative of three independent experiments. We have mentioned this in the Fig. legend in the revised manuscript. Although hNPCs was susceptible to ZIKV infection, the final titers of progeny ZIKV are far from other culture cells like Vero or BHK-21 cells; this is consistent with our previous findings [10] and others [11].

9. Line 145 (Fig. 3F): I would not label this axis “positive rate” but “% of infection” or something similar.

Reply: Corrected as the reviewer’s suggestion.

10. Line 147. Instead of enhancing replication, it could also enhance binding or entry, please show that this is not altered.

Reply: Thanks for the insightful suggestion. As suggested, we performed the binding

and entry assay for WT and K101R. There is no obvious difference in terms of the virus-cell binding capacity and entry efficiency between WT and K101R. The data were provided as new **Fig. 7D and 7E** in the revised manuscript.

11. Line 158. Please include data on infectious virus titers.

Reply: As suggested, we further tested infectious virus titers in brains of mice i.c. infected with WT, K101R or MR766. New data were provided as new **Fig. 3A-3C** in the revised manuscript.

12. Line 159. Please explain why this difference is only observed on day 9?

Reply: Thanks for the comments. To further validate our finding, we performed another round of animal experiments, and the viral RNA and infectious viral particles in brains of suckling mice i.c. inoculated with WT, K101R and MR766 were determined on days 7, 9 and 11. We once again observed that K101R showed significantly increased in viral RNA on day 9 post inoculation (new **Fig. 3B**). Moreover, the titer of infectious viral particles was significantly increased in brains of K101R-inoculated mice on days 7 and 9 post inoculation compared with WT (new **Fig. 3C**). The findings verify the K101R mutation at least confers WT a replication advantage in mouse brains until day 9 post inoculation. Notably, no noticeable difference in replication between K101R and WT was still observed on day 11 post inoculation. Viral replication in mouse brains might be affected by many factors such as animal age, viral dose, injection routes, and individual host responses. Similar findings have been well documented in previous findings [12-14].

13. Line 161: What do the brains of day 11 mice look like?

Reply: Since no significant difference between WT and K101R was observed in viral loads in mice on day 11 post inoculation, we did not perform the tissue immunofluorescent staining to examine the viral proteins in the brains of animals. There is no visible difference in brain morphology between the two groups.

14. Line 166-170. What is the rationale for performing the transcriptomics on day 11, while the difference in replication is observed on day 9?

Reply: Thanks for the comment. As transcriptomic analysis were performed to identify the differentiated genes, the time points were chosen 2 days after when differentiated viral replication was detected to expand the host response. As expected, distinction in transcriptomic patterns were observed between the brains of mice infected with WT or K101R on day 11 post inoculation. The detailed description of transcriptomic results was provided in the revised manuscript (**lines 192-219**).

15. Line 166-198. Please include analysis of unique and common genes. The genes unique to either virus may identify virus specific responses.

Reply: Thanks for the valuable suggestion. We made a detailed analysis of unique and common genes induced by WT and K101R in the results and discussion sections in the revised manuscript (lines 192-219, 320-346).

Line 206: I don't really understand this rationale – why did you assume that K101R would abolish RNA binding capability? Wouldn't this hamper RNA packaging therefore making the virus less virulent, which is not what you claim with the rest of your data?

Reply: Thanks for the comment. We feel sorry for the misleading remarks, probably due to the use of “abolish”. Flavivirus capsid protein is a typical multiple function proteins, including direct binding to viral RNA within the virion. As RNA binding activities of the capsid protein have been reported to be likely mediated through nonspecific hydrophobic or charged interactions [15]. To rule out the possibility that K101R directly influence the RNA binding ability, RNA binding comparison between the mutated C proteins were performed. Further discussion about the effect of the mutation on the RNA binding ability of C protein was also amended in the revised manuscript (lines 285-290).

16. Line 216 (Fig. 6D): You showed increased replication of K101R in the BHK-21 cells, and here it looks like there is indeed more virus present as shown by the intensity of the envelope. Could the ratio be misrepresented by there simply being a higher input of virus? Why not do western blot on a standardized virus input? Also, the ratio of immature to mature looks quite different between FSS13025 vs WT, which is indicated in panel E. Why does this difference exist? Additionally, please correct “Envelope” to “Envelope”

Reply: We are apologized for the confusion. During the viral replication, the immature anchored C protein (1-122 amino acids) is cleaved to release a mature capsid protein (1-104 amino acids). The mature capsid protein then assembles with the viral RNA genome to facilitate the packaging into the virus particle. To examine the maturation efficiency of C protein, we quantified the immature capsid (about 15 kD) and mature capsid (about 12 kD) by calculate the gray values of the C protein as previously described [16]. As shown in Fig. 6F and 6G, the ratio of the mature to immature C proteins is higher in cells infected with K101R compared to WT. Although there is a slight difference in the intensity of the E protein between WT- or K101R-infected cells, we believe that ratio of different types of C protein accurately reflects its maturation process, supporting the conclusion that maturation efficiency of the C protein is higher

in cells infected with K101R than WT. With regards to maturation pattern of C protein of the FSS13025 and WT strains, the ratio of mature to immature capsid protein was 58% and 80%, respectively. Similar assays have been well performed in previous papers [16] in the field.

“Envolope” has been corrected to “Envelope”. Sorry for the typo.

17. Line 233. In the main text you state “all” ZIKV strains and not selected ZIKV strains.

Reply: As suggested by the reviewer, we have corrected this sentence accordingly (lines 255-258)

18. Line 246: Does the residue at position 106 have a similar function as you have observed with your C protein mutation?

Reply: Thanks for the insightful comment. In Yu’s report, mechanical experiments showed the T106A substitution of the C protein of ZIKV rendered the C a preferred substrate for the NS2B-NS3 protease, thereby promoting the maturation of structural proteins and the formation of infectious viral particles [17]. Considering that both positions 101 and 106 of the C protein are immediately next to the NS2B-NS3 protease cleavage site, it is reasonable to speculate residue substitutions at these positions could potentially affect the mature process of the C protein. Indeed, by calculating the ratio of mature and immature protein, our study provided the evidence that position 101 has a similar influence on the mature process of the C protein of ZIKV. Both our study and Yu's report shed light on the importance of amino acids located immediately next to the NS2B-NS3 protease cleavage site in the biological functions of ZIKV. We also added this text to the discussion section accordingly (lines 295-304) in the revised manuscript.

19. Line 247. What is the effect of the K101R mutation in mosquitoes?

Reply: To address this important comment, we perform growth curve of K101R and WT in mosquito cell lines. And the results showed that K101R showed enhanced replication compared with the WT virus (Fig. 3G and 3H). Due to facility limitation, we are not able to further validate this result in live mosquitoes. We believe this will be an interesting point to be investigated in the future.

20. Line 263. How does the increase in production of mature C protein result in increased replication? This suggests that the availability of mature C is the limiting factor in replication?

Reply: We thank the reviewer for raising this important point. The primary function of

flavivirus C protein is to encapsulate the viral genome. The immature form of C protein has a C-terminal membrane-spanning anchor, which is cleaved by the viral NS2B-NS3 protease, permitting the release of the mature C protein at the cytoplasmic side. Subsequently, mature C proteins recruit the newly synthesized viral genome to form the nucleocapsid complex. Thus, an increased maturation efficiency of C protein contributes to the assembly of viruses, leading to enhanced virus replication. Multiple studies have shown that large deletions throughout, or large insertions at the C terminus yield viable but attenuated virions [18-20]. Additionally, ZIKV C protein has been reported to subvert the type I IFN response [22]. Therefore, enhanced production of mature C protein may also contribute to suppression of antiviral responses in hosts. Indeed, our findings show the replication of K101R was significantly higher than that of WT in mosquito C6/36 and Aag2 cell lines (**Fig. 3G and 3H**). Collectively, the maturity of C protein contributes to viral genome replication, virion assembly as well as host antiviral responses.

21. Line 275: ‘‘Might do not change’’ does not make sense.

Reply: We apologize for the inaccurate statement. ‘‘Might do not change’’ has been deleted in the revised manuscript.

22. Line 282. Same question as line 263. Does neurovirulence depend on replication or host response, or both?

Reply: Neurovirulence refers to the ability of a viral infection to cause central nervous system (CNS) pathology. It is generally dependent on the virus replication in CNS and/or host immune responses. As responded above, the C protein play roles in virus assembly and disturbing the immune responses in hosts. Therefore, enhanced neurovirulence may be attributed to the increased replication of virus and/or more efficient suppression of the host responses.

23. Line 345. How was this analysis performed?

Reply: We apologize that this information was missing in the original manuscript, and we have now added this to the ‘‘Materials and methods’’ section in the revised manuscript (**lines 410-437**).

24. Line 348. Why were these 2 additional mutations included?

Reply: As FSS13025 strain is isolated in 2010 in Cambodia. We and others have demonstrated the contemporary Asian isolates contained two amino acid substitutions S139N and M2634V that are evolutionary important [23]. So the infectious clone that

carry both S139N and M2634V were used as backbone in our present study.

25. Line 399. What volume was used for inoculation? And please include a mock control for effect of inoculation.

Reply: We inoculated i.c. the suckling mice with 30 uL of PBS, and this detail has been included in Materials and Methods in revision. No mortality and morbidity were observed in mice during the infection course. New data were provided as **Fig. S1C and Fig. 2E** in the revised manuscript.

Reviewer #2 (Remarks to the Author):

In their manuscript “A single amino acid substitution in the capsid protein of Zika virus contributes to the African lineage-specific virulence phenotypes,” Song et al. identify a specific mutation (K101R) to explain the virulence of African lineage Zika virus. Through phylogenetic and sequence analysis, they find distinguishing mutations that could contribute to the virulence. They generate point mutations in Zika infectious clones and find the K101R in C protein increases viral PFU in cell lines and RNA in neonatal mice brains. The K101R has increased C protein cleavage of the K101R mutant compared to WT, which could be facilitating the maturation of structural proteins and formation of infectious viral particles. This is an interesting study but focuses exclusively on African-lineage neurovirulence; it is unclear how it relates to neurovirulence of i.e. the Puerto Rico strain that does not contain the mutation.

Reply: Thanks for the encouraging comments.

Specific comments:

1. Please include MR766 in the infections presented in Fig. 2 and 3 in order to fully compare the virulence of the K101R to both WT and MR766.

Reply: Thanks for the valuable suggestion. New data on replication of MR766 in cell lines and brains of suckling mice were provided as new **Fig. 3A-3D, 3G, and 3H** in the revised manuscript.

2. Please quantify the percent of cells infected with the different Zika viruses and the GFP MFI. It appears that there are a lot more infected cells (and could be brighter GFP signal) in the K101R compared to WT.

Reply: We appreciate the reviewer’s insightful observation. As suggested, we measured the percent of cells infected with recovered ZIKVs. The virus infectivity results showed the K101R virus resulted in higher percentages of infected cells, with an average of 80%

than the WT virus, that displayed about 45% (shown below). This finding is consistent to the observation that K101R resulted in higher levels of viral RNA and infectious virus particles in BHK-21 cells than that of WT (**Fig. 3A and 3B** in the revised manuscript).

3. Are there differences in the temperature differences or amount of viral RNA in the mice infected in Fig. 2E?

Reply: Thanks for the comments. The experiment shown in Fig. 2E was performed to screen mutations with possible capacity of playing a role in the neurovirulence of ZIKV for mice. Therefore, we just observed mortality of suckling mice i.c. inoculated with all the recovered mutant viruses, and no other assays were performed. Mouse body temperature has been established to link to ZIKV infection, and we didn't measure in our original experiments. Detailed comparison between K101R and the WT virus was tested in following assays shown in **Figs 3 and 4**.

4. Please provide viral RNA quantification of the infected cells in Fig. 3.

Reply: Thanks for the valuable suggestion. As suggested, we examined comprehensively the replication in multiple cell lines including BHK-21, Vero, C6/36 and Aag2 cells. Viral RNA and infectious virus particles were quantified using RT-qPCR and plaque assay, respectively. These results were included as new **Fig. 3A-3D, 3G, and 3H** in the revised manuscript.

5. Is the mouse experiment in Fig. 4 independent of the one presented in Fig. 2? Is there survival/temperature data for Fig. 4? What about PFUs in the brain?

Reply: We apologized for this confusion. In the original manuscript, the mouse experiment in Fig. 4 was independent of the animal experiment in Fig. 2E. To provide solid data on neurovirulence of K101R for mice, neurovirulence LD₅₀s of WT, K101R and MR766 were calculated. Further, viral replication in mouse brains was examined by quantifying viral RNA and infectious viral particles. These new data were included

in the new **Fig. 4A-4C** in the revised manuscript.

6. Please include statistics for Fig. 6D/E.

Reply: As suggested by the reviewer, statistics was added for the new **Fig. 6G** in the revised manuscript.

7. Please discuss the possibility of the effects of the K101R mutation on affecting posttranslational modifications of the C protein.

Reply: Thanks for this insightful suggestion. A discussion on the K101R substitution on the post translational modification of the C protein was included in the discussion section in the revised manuscript (**lines 305-310**).

8. The Puerto Rico (DB-ZIKV966) lineage from 2016 (Fig. 1) is known for its increased virulence and neuropathogenesis, but does not have the K101R mutation. Please discuss how the K101R model fits with the virulence of the PR virus.

Reply: This is a very good point. Puerto Rico (DB-ZIKV966), as well as some other Asian strains, has been well documented to have increased virulence. The virulence phenotype is a relative concept that varies depending on the specific context and circumstances. Among all factors, a panel of amino acid substitutions including K101R reported in our study "contribute" but not "determine" to the virulence phenotype observed. Correspondingly, we have recently confirmed that E21K and S139N mutations in prM also contribute to the enhanced neurovirulence of ZIKV [24, 25]. We have now noted this point in the discussion in the revised manuscript (**lines 272-275**).

Reviewer #3 (Remarks to the Author):

In this article, the authors investigated the determinant(s) of increased virulence of the historical isolate of ZIKV strain of African lineage (MR766) as compared to a contemporary isolate (FSS13025) of Asian lineage. Phylogenetic analysis revealed six African lineage-specific residues in the structural proteins that were different from the Asian lineage virus. Using an infectious clone of FSS13022, individual mutations were introduced into the clone and viruses were characterized for their growth in vitro. Of all the mutants examined, the K101R mutant virus (having an arginine in place of lysine in the C protein) was shown to possess enhanced replication phenotype and exhibited increase neurovirulence

properties compared to the wild-type (WT) infectious-clone derived FSS13022 virus. The authors performed RNA-seq analysis of the brains of animals infected with the mutant and WT virus to determine the genes that are up-regulated or down-regulated in response to virus infections. Further, the authors demonstrated that K101R mutation leads to increased production of the mature C protein. Overall, the authors conclude that K101R substitution in C protein is a key determinant of lineage-specific virulence of ZIKV.

Although the results are interesting, the studies are very limited in nature. The paper lacks any demonstration of conceptual advance in our understanding of ZIKV replication, virulence, and pathogenesis. Furthermore, the studies are neither novel nor impactful. I believe the paper does not meet the quality and novelty that are expected of papers published in Nature Communication.

Reply: Thanks for the comment. ZIKV has been well recognized as one of the most important and mysterious viruses worldwide. The identification and characterization of neurovirulence determinants represents as the most critical scientific question to be answered. Since the identification of A982V-NS1 and S139N-prM substitutions in **Nature** and **Science** [25, 26], some additional mutations in capsid protein, such as T106A and H41R, were recently identified with unique biological functions in **Cell Stem Cell** and **PNAS** [17, 27]. Especially, a recent paper published in **Nat Commun.** 2021 observed the enhanced neurovirulence phenotype of African ZIKV [1]. Our present finding provides a direct answer to this fundamental question. We and others have noticed that the African lineage ZIKVs displayed enhanced virulence phenotypes in multiple cell and mouse models in comparison with the ongoing Asian lineage ZIKV. The underlying mechanism accounting for this unusual phenotype requires a reasonable clarification. Our study discovered a lineage-specific virulence determinant, K101R, unique to the African lineage ZIKVs. Therefore, we believe that this report not only expands the understanding of ZIKV pathogenesis and epidemiology in associated with ZIKV lineages, but also provides a novel target for live-attenuated vaccine development as well as molecular surveillance.

Specific Comments:

1. The title of the paper is misleading as it conveys that only a single mutation in the C protein is responsible for the higher virulence phenotype of African lineage virus. This is certainly not the case. Many studies in the literature have identified other mutations and proteins that contribute the virulence.

Reply: Thanks for the interesting comment. Our original title, which reads, ‘A single amino acid substitution in the capsid protein of Zika virus contributes to the African lineage-specific virulence phenotypes.’ The term ‘contribute’ denotes ‘a cause factor’

rather than the term ‘is responsible for’, which indicates ‘the definite cause of sth’. Therefore, our study claimed the position 101 of C protein is one of the key residues critical for the higher virulence phenotype of African lineages, this doesn't exclude the role of other critical residues that have been identified previously, and we have cited and credited these previous significant contributions in our original and present MS.

2. Fig. 1 presents studies that only reconfirm the existing data in the literature. No novel or new data are presented here. Also, the sequence alignment and phylogenetic analysis used sequences of only about 30 or so (out of hundreds available in database) isolates.

Reply: Thanks for the comments. First, we agree with the reviewer that the virulence phenotypes between Asian and African lineages have been documented previously; Although we used different representative strains, the conclusion was similar to previous finding, so we moved original Fig. 1A-1C into Supplemental material as new Fig. S1. Secondly, our evolutionally analysis was based all ZIKV genome available on the GenBank, only representative strains were listed in the Fig. due to Fig. space limitation. We have incorporated a detailed account of the methodology used for selecting ZIKV genome sequences and executing sequence alignment and phylogenetic analyses in the "Materials and Methods" section ((**lines 410-437**)).

3. Figs 1C and 2E: It is not clear why the authors used two different doses of the virus for i.c. inoculation for neurovirulence studies, particularly when just 1 PFU of MR766 virus was sufficient to kill all mice.

Reply: Thanks for the comments. The selection of dose was chosen based on our experimental design and protocols. Fig. 1C was designed to compare the neurovirulence phenotype between FSS13025 and MR766. Fig. 2E was devised with the aim of identifying the mutants that are different from the WT virus, and the dose was chosen based on our in-house experimental protocol and animal use rules.

4. Fig. 3: The growth differences of WT and K101R viruses in most cases appear to be marginal (although statistically significant).

Reply: To address the reviewer's concern, we comprehensively tested the replication of viruses, including K101R, WT and MR766, in mammalian and mosquito cell lines as well as in mouse brains. Growth curve analysis in cells showed K101R replicated more efficiently than WT in terms of viral RNAs and infectious viral particles, despite of a difference of less than 1 log (new **Fig. 3A-3D, 3G, and 3H** in the revised manuscript). Notably, K101R showed a significantly increased accumulation of viral RNA and infectious particles in mouse brains in comparison with WT, with a difference

about 1-2 logs (**new Fig. 4** in the revised manuscript). Consistent with this, the LD₅₀ of K101R for suckling mice showed about 35-fold reduction in comparison with that of WT (**new Fig. 4A** in the revised manuscript). Therefore, increased replication of K101R in cells holds biological relevance even with a difference of less than 1 log.

5. Fig. 5: The authors conducted transcriptomic analysis to “... further investigate the underlying mechanism...” (line 167) but following the analysis, the results were presented in a very superficial manner. It is difficult to determine what is specifically up-regulated and what is down-regulated in K101R virus as compared to WT virus. How the differentially expressed genes affect the virulence phenotypes of the two viruses are not discussed. Since the authors used brains of animals infected with the two viruses for the analysis, it would have been interesting to examine genes that are involved specifically in neurogenesis, and/or those that are involved in microcephaly. Overall, these studies and the results, as described here, appear to be simply a fishing expedition with no real hypothesis and clear cut analysis of the data.

Reply: Thanks for the insightful comment. We agree with the reviewer that we can not directly make sure the specially regulated genes, that's also not our major purposes. RNA-seq data help us understand the comprehensive host response to ZIKV infection. Indeed, we can't simply link the differentially expressed genes to the observed virulence phenotypes considering the complexity of the *in vivo* system. Similar to most papers in the field, the differentially expressed genes and pathways, probably, contribute to the observed phenotype. Thus, the transcriptomic data should be collectively combined with our biochemical, virological, and structural data, as all these factors contribute to the final outcome. Whatever, we further made an in-deep analysis of unique and common genes induced by WT and K101R. The detailed description of transcriptomic results was also updated as new **Fig. 5C-5G** provided in the revised manuscript (**lines 186-219**).

6. Fig. 6: To their credit, the authors have demonstrated that the K101R mutation results in production of higher levels of mature C protein. But it is not clear how this occurs as both of these residues are positively charged. The authors postulate that “..K101R substitution is likely to change the dibasic motif at the P1 and P2 position....” (line 271). Is it possible that since R is more positively charged as compared to K, the cleavage efficiency is enhanced?

Reply: Thanks for this insightful comment. To describe more accurately, we rewrote the description of effect of the K101R substitution on the charge in the revised manuscript, and have discussed this point in the Discussion (**lines 281-304**).

References in this letter

1. Aubry, F., et al., *Recent African strains of Zika virus display higher transmissibility and fetal pathogenicity than Asian strains*. Nat Commun, 2021. **12**(1): p. 916.
2. Jaeger, A.S., et al., *Zika viruses of African and Asian lineages cause fetal harm in a mouse model of vertical transmission*. PLoS Negl Trop Dis, 2019. **13**(4): p. e0007343.
3. Crooks, C.M., et al., *African-Lineage Zika Virus Replication Dynamics and Maternal-Fetal Interface Infection in Pregnant Rhesus Macaques*. J Virol, 2021. **95**(16): p. e0222020.
4. Koenig, M.R., et al., *Infection of the maternal-fetal interface and vertical transmission following low-dose inoculation of pregnant rhesus macaques (*Macaca mulatta*) with an African-lineage Zika virus*. PLoS One, 2023. **18**(5): p. e0284964.
5. Newman, C.M., et al., *Early Embryonic Loss Following Intravaginal Zika Virus Challenge in Rhesus Macaques*. Front Immunol, 2021. **12**: p. 686437.
6. Nakayama, E., et al., *Neuroinvasiveness of the MR766 strain of Zika virus in IFNAR^{-/-} mice maps to prM residues conserved amongst African genotype viruses*. PLoS Pathog, 2021. **17**(7): p. e1009788.
7. Simonin, Y., et al., *Differential virulence between Asian and African lineages of Zika virus*. PLoS Negl Trop Dis, 2017. **11**(9): p. e0005821.
8. Shao, Q., et al., *The African Zika virus MR-766 is more virulent and causes more severe brain damage than current Asian lineage and dengue virus*. Development, 2017. **144**(22): p. 4114-4124.
9. Tripathi, S., et al., *A novel Zika virus mouse model reveals strain specific differences in virus pathogenesis and host inflammatory immune responses*. PLoS Pathog, 2017. **13**(3): p. e1006258.
10. Chen, X., et al., *Zika virus RNA structure controls its unique neurotropism by bipartite binding to Musashi-1*. Nat Commun, 2023. **14**(1): p. 1134.
11. Anfasa, F., et al., *Phenotypic Differences between Asian and African Lineage Zika Viruses in Human Neural Progenitor Cells*. mSphere, 2017. **2**(4).
12. Annamalai, A.S., et al., *Zika Virus Encoding Nonglycosylated Envelope Protein Is Attenuated and Defective in Neuroinvasion*. J Virol, 2017. **91**(23).
13. Daniels, B.P., et al., *The Nucleotide Sensor ZBP1 and Kinase RIPK3 Induce the Enzyme IRG1 to Promote an Antiviral Metabolic State in Neurons*. Immunity, 2019. **50**(1): p. 64-76 e4.
14. Gorman, M.J., et al., *An Immunocompetent Mouse Model of Zika Virus Infection*. Cell Host Microbe, 2018. **23**(5): p. 672-685 e6.
15. Barnard, T.R., et al., *Molecular Determinants of Flavivirus Virion Assembly*. Trends Biochem Sci, 2021. **46**(5): p. 378-390.
16. Tan, T.Y., et al., *Capsid protein structure in Zika virus reveals the flavivirus assembly process*. Nat Commun, 2020. **11**(1): p. 895.
17. Yu, X., et al., *A mutation-mediated evolutionary adaptation of Zika virus in mosquito and mammalian host*. Proc Natl Acad Sci U S A, 2021. **118**(42).
18. Kofler, R.M., F.X. Heinz, and C.W. Mandl, *Capsid protein C of tick-borne encephalitis virus*

- tolerates large internal deletions and is a favorable target for attenuation of virulence.* J Virol, 2002. **76**(7): p. 3534-43.
19. Patkar, C.G., et al., *Functional requirements of the yellow fever virus capsid protein.* J Virol, 2007. **81**(12): p. 6471-81.
 20. Schrauf, S., et al., *Extension of flavivirus protein C differentially affects early RNA synthesis and growth in mammalian and arthropod host cells.* J Virol, 2009. **83**(21): p. 11201-10.
 21. Samuel, G.H., et al., *Yellow fever virus capsid protein is a potent suppressor of RNA silencing that binds double-stranded RNA.* Proc Natl Acad Sci U S A, 2016. **113**(48): p. 13863-13868.
 22. Airo, A.M., et al., *Flavivirus Capsid Proteins Inhibit the Interferon Response.* Viruses, 2022. **14**(5).
 23. Liu, Z.Y., W.F. Shi, and C.F. Qin, *The evolution of Zika virus from Asia to the Americas.* Nat Rev Microbiol, 2019. **17**(3): p. 131-139.
 24. He, M.J., et al., *Key Residue in the Precursor Region of M Protein Contributes to the Neurovirulence and Neuroinvasiveness of the African Lineage of Zika Virus.* J Virol, 2023. **97**(3): p. e0180122.
 25. Yuan, L., et al., *A single mutation in the prM protein of Zika virus contributes to fetal microcephaly.* Science, 2017. **358**(6365): p. 933-936.
 26. Liu, Y., et al., *Evolutionary enhancement of Zika virus infectivity in Aedes aegypti mosquitoes.* Nature, 2017. **545**(7655): p. 482-486.
 27. Zeng, J., et al., *The Zika Virus Capsid Disrupts Corticogenesis by Suppressing Dicer Activity and miRNA Biogenesis.* Cell Stem Cell, 2020. **27**(4): p. 618-632 e9.

REVIEWERS' COMMENTS

Reviewer #1 (Remarks to the Author):

Overall, the authors have addressed this reviewers comments. However, given the concerns about how representative the MR766 strain is as an African lineage. The authors should focus the 101 mutation as a mechanistic question, rather than it's biological relevance in the field. As such, I would recommend changing the title to something like: "A single amino acid substitution in the capsid protein of Zika virus contributes to a neurovirulent phenotype"

And leave the link to infection in patients and potential difference between African and Asian lineages to the discussion.